# Large-area, untethered, metamorphic, and omnidirectionally stretchable multiplexing self-powered triboelectric skins

Beibei Shao[1,2,7], Ming-Han Lu[3,7], Tai-Chen Wu[3], Wei-Chen Peng[3], Tien-Yu Ko[3], Yung-Chi Hsiao[3], Jiann-Yeu Chen[4], Baoquan Sun [1,2,5] ✉, Ruiyuan Liu [1,2] ✉ & Ying-Chih Lai [3,4,6] ✉

Large-area metamorphic stretchable sensor networks are desirable in haptic sensing and next-generation electronics. Triboelectric nanogenerator-based self-powered tactile sensors in single-electrode mode constitute one of the best solutions with ideal attributes. However, their large-area multiplexing utilizations are restricted by severe misrecognition between sensing nodes and high-density internal circuits. Here, we provide an electrical signal shielding strategy delivering a large-area multiplexing self-powered untethered triboelectric electronic skin (UTE-skin) with an ultralow misrecognition rate (0.20%). An omnidirectionally stretchable carbon black-Ecoflex composite-based shielding layer is developed to effectively attenuate electrostatic interference from wirings, guaranteeing low-level noise in sensing matrices. UTE-skin operates reliably under 100% uniaxial, 100% biaxial, and 400% isotropic strains, achieving high-quality pressure imaging and multi-touch real-time visualization. Smart gloves for tactile recognition, intelligent insoles for gait analysis, and deformable human-machine interfaces are demonstrated. This work signifies a substantial breakthrough in haptic sensing, offering solutions for the previously challenging issue of large-area multiplexing sensing arrays.

The skin, the largest organ of the human body, possesses many attractive attributes, including flexibility, stretchability, and the ability to self-heal after injury[1–4]. It serves as a fundamental interface to communicate human emotion and as a key receptor to external stimuli including touch, pressure, dampness, and temperature[5–7]. Particularly, the skin has a self-active perceiving capability[8,9]. Electronics mimicking skin are of interest and bring about various emerging technologies, ranging from wearable/stretchable/biomedical/healthcare electronics and human-device interfaces to robotic and prosthetic skins. Conventional skin-like sensory devices are designed via passive technologies, such as capacitive, resistive, and optical[10–14]. Such passive devices suffer from complicated device structures, scant flexibility, and limited

[1]Soochow Institute of Energy and Material Innovations, Key Laboratory for Advanced Carbon Materials and Wearable Energy Technologies of Jiangsu Province, Institute of Functional Nano & Soft Materials (FUNSOM) and College of Energy, Soochow University, Suzhou 215006, PR China. [2]Jiangsu Key Laboratory of Advanced Negative Carbon Technologies, Soochow University, Suzhou 215123, PR China. [3]Department of Materials Science and Engineering, National Chung Hsing University, Taichung 40227, Taiwan. [4]Innovation and Development Center of Sustainable Agriculture, i-Center for Advanced Science and Technology, National Chung Hsing University, Taichung 40227, Taiwan. [5]Macau Institute of Materials Science and Engineering MUST-SUDA Joint Research Center for Advanced Functional Materials Macau University of Science and Technology Macau, 999078 Macao, PR China. [6]Department of Physics, National Chung Hsing University, Taichung 40227, Taiwan. [7]These authors contributed equally: Beibei Shao, Ming-Han Lu. ✉e-mail: bqsun@suda.edu.cn; ryliu@suda.edu.cn; yclai@nchu.edu.tw

scalability, and require continuous electrical or optical signals to drive their operation[15,16]. In this framework, to realize a large-area and multiplexing metamorphic sensing matrix, a large voltage (1–10 V) for each stretchable sensor and a large amount of continuous power dissipation become critical issues[17–19].

Compared with passive transducer technologies[5,20,21], triboelectric nanogenerators (TENGs), derived from the triboelectrification effect and electrostatic induction coupling, have garnered increased attention owing to their unique favorable attributes, including self-powered operation, material and configuration diversity, and superior sensitivity[22–25]. When two materials come in contact, the TENG can transform mechanical energy into electrical signals with a relatively high amplitude, enabling its high-precision tactile recognition and pressure mapping[17,26,27]. For example, using elastomeric polydimethylsiloxane (PDMS) as electrification layers and patterned Ag nanofibers as electrodes and circuit connections, a stretchable (>100% strain) and self-powered 8 × 8 triboelectric tactile sensing array was achieved using a cross-locating technology[28]. Moreover, a skin-like TENG sensing matrix with a 3 × 3 array has been developed by using PDMS and stretchable thermoplastic polyurethane/Ag nanowires (AgNWs) electrode[29]. However, a prevailing and fatal issue for single-electrode triboelectric sensor arrays is the contradiction between the multiplexing design and the severe misrecognition, which primarily derives from the electrical signal interferences between the sensing nodes and the internal circuits[30–33]. This issue arises from complicated high-density electrode wiring design upon multiplexing and large-area detection. When high-density wirings receive external force/touch, they inevitably produce pronounced electrical signals owing to the combined effects of triboelectrification and electrostatic induction, resulting in the incorrect identification of real signals triggered by sensing pixels[34–37]. Such misrecognition conveys false or misleading messages about the physical stimuli, drastically sacrificing the sensing fidelity and hindering the applications of the TENG sensing matrix[38]. Thus, effectively preventing or restraining misrecognition and realizing accurate tactile detection in multiplexing and large-area sensing arrays is extremely crucial for practical uses.

Extensive efforts have been made to eliminate the misrecognition issue by erecting delicate configurations of sensing nodes and electrodes, i.e., constructing intersections between row and column electrode lines that possess matching patterns[39], setting height differences between sensor cells and electrodes[40], or building sensing units with patterned subdivisions[41]. Despite impressive advances, integrating particular patterns complicates the design and construction of data acquisition. Moreover, the mismatch between the elastic moduli of the separate functional layers causes concentrated strains at the interfaces, which severely compromises the robustness of the entire system. An alternative approach is using conductive shield layers such as nickel-deposited fabric and spray-coated AgNWs to suppress misrecognition, which boosts object-sensing resolution[39,42]. However, previous studies only achieved flexible TENG sensing arrays. The development of large-area multiplexing sensing arrays with excellent stretchability and elasticity is challenging because of the requirements for the corresponding mechanical deformability of the sensing nodes, conducting wires, shield layers, and triboelectric layers. In particular, for stretchable and elastic large-area multiplexing sensing networks, misrecognition is even more intractable. Modulus matching of each functional layer is another critical issue.

Here, we propose a large-scale untethered multiplexing TENG tactile sensing array, termed untethered triboelectric electronic skin (e-skin) (UTE-skin), with the combined attributes of self-powered sensing, large stretchability and elasticity, robustness, and an ultra-low misrecognition rate of 0.20%. The entirely and inherently stretchable UTE-skin (4 × 4 pixels in a 25 cm × 25 cm area) working in the single-electrode mode is made of highly compliant electronic materials, including elastomeric Ecoflex as a stretchable triboelectric

layer and matrix, elastic carbon black-doped Ecoflex as a shielding layer, electrodes, and electrical interconnect. Owing to the electrostatic interference screening effect, the elastic composite-based shield layer can effectively eliminate the misrecognition between sensing nodes and internal connecting wiring, guaranteeing the stability, accuracy, and repeatability of the e-skin. The device also worked stably even under 100% uniaxial, 100% biaxial, and 400% isotropic tensile strains. Such a newly designed UTE-skin can provide clear pressure imaging and real-time visualization of multiple-point touch. UTE-skins were demonstrated for use as intelligent gloves for gesture recognition and smart insoles for gait detection. In parallel, they permit the buildup of system-level sensing platforms toward immersion-reinforced training involving human-machine interfaces, such as a real-time spherical game controller and wearable smartphone keypad. The proposed scheme and device are promising cornerstones of scalable untethered multiplexing self-powered e-skins, which are highly desired in practical applications, including haptics, human-device interfaces, medical care/assistance, and human-like/robotic perception.

## Results
### Design of large-area UTE-skins

A large-area, stretchable, and untethered multiplexing self-powered e-skin with a 4 × 4 TENG tactile sensing array was constructed layer-by-layer by heterogeneously integrating all the elastomeric films into a thin layout, as illustrated in Fig. 1a. First, a carbon black-doped Ecoflex shielding layer with a thickness of 1.0 mm was smeared from its solution over a flat acrylic mold and ground to screen for electrostatic interference. Specifically, the shielding layer was built on a conductive composite of carbon black percolation networks and Ecoflex silicone rubber, which boasts exemplary electrical properties owing to the robust interfacial bonding between the carbon black and Ecoflex matrices[43]. After fully curing, the shielding layer was cut into a 4 × 4 sensing array. Pristine liquid Ecoflex was then poured onto the patterned shielding layer as the triboelectric layer (thickness of 1.0 mm). Ecoflex was selected as the triboelectric material because of its high tendency to collect electrons from human skin and other materials, and its outstanding elasticity and durability[44,45]. Another acrylic mold was tailored into a 4 × 4 array, and the corresponding interconnects were laminated onto the as-prepared composite in parallel. Then, the carbon black-doped Ecoflex was poured into its groove to construct elastic electrodes (i.e., the sensing nodes) and connecting circuits. After casting a layer of Ecoflex elastomer as the substrate, peeling off the entire device from the acrylic mold, and turning it upside down, the UTE-skin with three-layer lamination (each layer was 1.0 mm) was ultimately completed. In this work, the top Ecoflex layer functioned as the triboelectric layer in a 4 × 4 sensing array, which can create a continuous electrical output upon successive contact and separation with the human skin. Meanwhile, the bottom Ecoflex served as an ideal elastic substrate as well as encapsulating material that protected the elastic electrodes and connecting circuits from mechanical damage. It was also used as an underlying substance for placing onto other surfaces. In practice, there was no contact-and-separation between this side and the finger; therefore, it would not be triggered. The entire process and materials are room-temperature manufacturable, cost-effective, and industrially friendly, offering a powerful tool for developing diverse stretchable and conformable electronic patches where large-area multiplexing sensing matrices are required.

Figure 1b presents an enlarged view of the UTE-skin platform for multiple-point tactile sensing in the form of a 16-channel sensing patch. The manufacturing method, dimensions, and spacing were identical for each node. Each TENG sensing unit was operated in single-electrode mode[46]. Notably, the accuracy of a typical single-electrode TENG sensor array is compromised by the issue of misrecognition

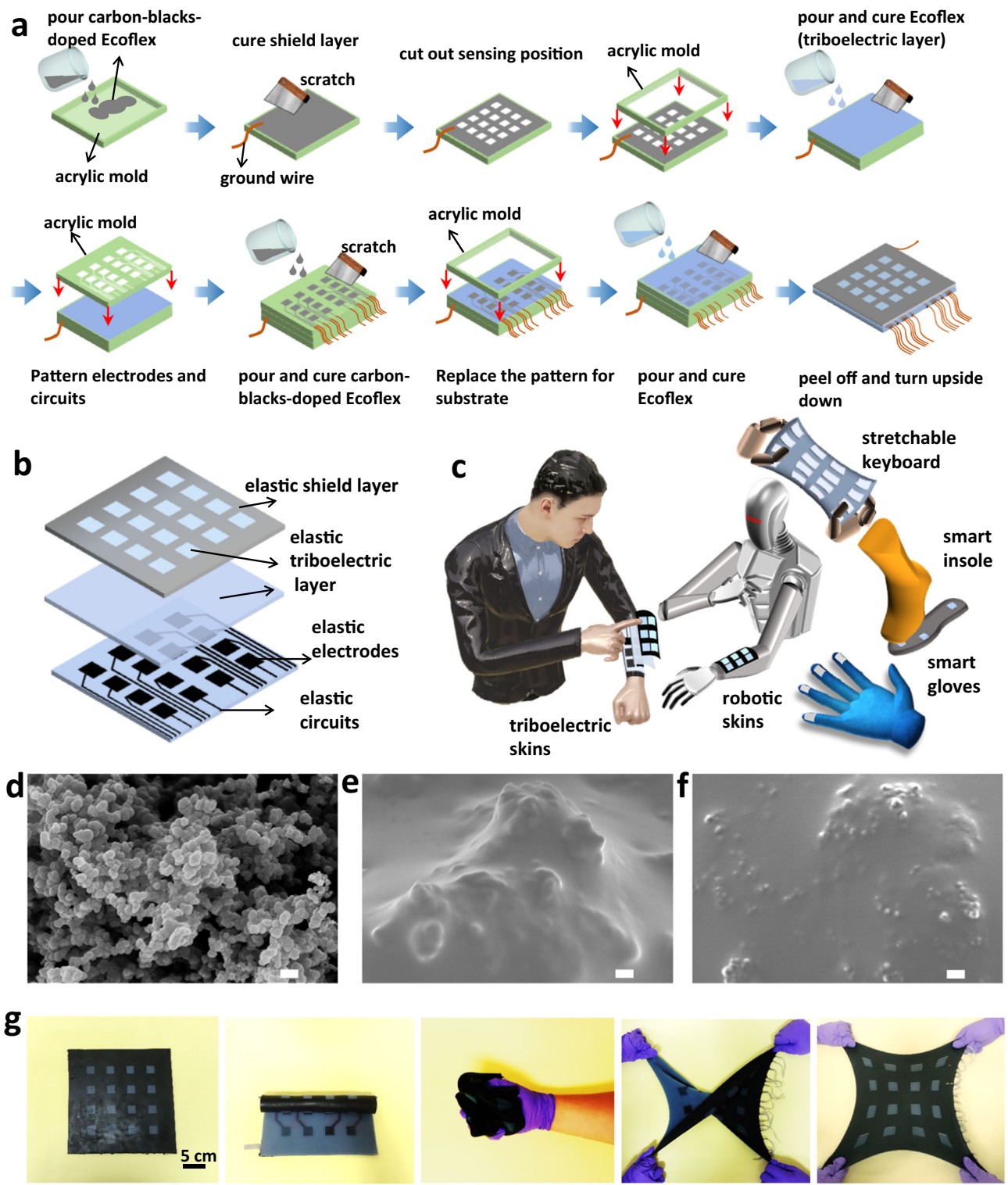

**Fig. 1 | Schematic design of the untethered triboelectric electronic skin (UTE-skin). a** Schematic illustration of the fabrication process for the stretchable UTE-Skin. **b** The exploded view of the UTE-skin architecture involving elastic circuits, elastic electrodes, elastic triboelectric layer, and elastic shielding layer. **c** The broad applications of the UTE-skin in robotic skins, stretchable keyboards, smart insole, and smart gloves. **d** Scanning electron microscope (SEM) image of carbon blacks. Scale bar: 100 nm. **e** Top-view and (**f**) cross-section SEM images of the conductive composite of carbon blacks and Ecoflex. Scale bar: 100 nm. Five independent experiments of (**d**–**f**) were performed with similar results. **g** Photographs of an as-fabricated triboelectric sensor array-based e-skin (left image) under folding, crumpling, twisting, and stretching, demonstrating its mechanical robustness.

between the sensing elements and connecting wires. Therefore, a top shielding layer was designed to eliminate misrecognition.

The intrinsically stretchable and deformable UTE-skin is entirely manufactured using compliant mechanical components. An Ecoflex elastomer with a high elastic limit (~800%) affords a compliant plateau for various building blocks[44], including the elastic circuits, electrodes, triboelectric layers, and shielding layers. The environmentally robust Ecoflex is biocompatible and durable for long-term functionalities[47];

therefore, it can be readily exploited as a skin-mounted device that accommodates various body motions. It also has a low Young's modulus (50-100 KPa), which is lower than that of soft skin tissues (0.1–2 MPa), thereby allowing for good biomechanical interactions with soft skin[1]. Moreover, the dexterity of the UTE-skin enables conformal contact with a wide range of nonplanar surfaces, including the rough and irregular surfaces of authentic objects. Accordingly, it can potentially be developed for a vast range of deformable haptic sensing devices, including electronic skins, robotic skins, stretchable keyboards, intelligent gloves, and smart insoles (Fig. 1c). Scanning electron microscopy (SEM) images show that the carbon black contained large quantities of nanoparticles arranged in a dense network (Fig. 1d). As shown in Fig. 1e, carbon blacks can be well embedded in the Ecoflex, and the composite contained large quantities of percolated carbon particles, which created conducting pathways in the composite. Meanwhile, the voids between carbon black particles enabled Ecoflex to penetrate the conducting networks and embed the particles. Figure 1f shows a cross-section SEM image of the composite, which also validated that the carbon blacks were embedded in the Ecoflex without delamination. Such morphological characteristics regarding conductive composites have also been validated for the previously reported carbon black and carbon nanotubes/Ecoflex composite[45], AgNWs/Ecoflex composite[30], Ag flakes/fluorine rubber[43], and coffee ground/Ecoflex composite[48]. Stress-strain curve of the Ecoflex substrate shows its fracture stress of 270% while the UTE-skin after three-layer lamination still demonstrates a stretchability of up to 200% (Supplementary Fig. 1). The stretchable UTE-skin meets the durability requirements of a wide range of wearable applications, accounting for less than 55% of the strain caused by typical human movement[15]. The UTE-skin also displays exceptional recovery according to the tensile stress-strain cycling tests of 100 times (100% strain) (Supplementary Fig. 2). The as-fabricated large-area soft and stretchable UTE-skin with dimensions of 25 cm × 25 cm is displayed in Fig. 1g. It exhibits excellent deformability and robustness to endure various extreme mechanical manipulations, including folding, crumpling, twisting, and stretching.

## Misrecognition reduction of UTE-skin with a shielding layer

Despite the superior potential of the TENG sensing array, a fatal issue arises from misrecognition, which signifies misidentified signals from high-density conductive wires. Figure 2a, b illustrate the working principles of the self-powered sensing node and screening electrostatic interference for the connecting wire, respectively. Here, we show that misrecognition issues can be effectively suppressed using a deformable carbon black-doped Ecoflex shielding layer.

As shown in Fig. 2a, the self-powered sensing node is a single-electrode-mode TENG sensor without a shielding layer. This involves a combination of contact triboelectrification and electrostatic induction[22,49]. According to the triboelectric series, Ecoflex boasts a higher electron affinity than human skin[50]. Thus, Ecoflex and human skin can be regarded as negatively charged and positively charged layers, respectively. Initially, the negative tribocharges on Ecoflex were electrically balanced by the positive charges in the internal electrode. As human skin approaches Ecoflex, the positive tribocharges on Ecoflex tend to be balanced by the negative tribocharges on the human skin. Consequently, free electrons can travel from the ground to the internal electrode to neutralize the inherent positive charges in the electrode, leading to a voltage/current signal to the external load. When the human skin begins to detach from the Ecoflex, the negative tribocharges on the Ecoflex can form a negative electric field again, which can repel the free electrons in the internal electrode to flow back to the ground, giving a reverse electrical signal. Successive contact and separation of the human skin and Ecoflex can create a continuous electrical output.

The elastic connecting wires and electrodes of the single-electrode TENG sensing nodes were all constructed using percolated carbon black-doped Ecoflex. Therefore, the absence of a shielding layer on the multiplexing TENG sensing array could cause misidentified signals when human skin or interacting objects are in contact with the connecting wires. Figure 2b depicts the mechanism of electrostatic interference screening using a grounded shielding layer on top of the connecting wires. The shielding layer comprises elastic conducting composites of Ecoflex and interlocked carbon black particles. The shielding layer screens the electrostatic charges generated on the connecting wires upon touching the human skin. Charge flowing between the ground and conducting wires would significantly be degraded, thereby resulting in misidentified electrical signals. Fortunately, the electrical properties of the shielding layer complement the sensing process of the sensor array, and the conductive percolating carbon black in the Ecoflex is imperative to eliminate the interference from surface triboelectrostatic induction and guarantee distinctive electrical signal outputs for accurate sensing functions. Here, we can readily modulate the conductivity of the elastic shielding layer by merely tailoring the carbon black content in the Ecoflex elastomer.

To further substantiate the proposed mechanism of electrical misrecognition annihilation by the shielding layer, we simulated the potential distribution of the TENG in four typical stages during the charged-skin-approaching process using finite element simulation (COMSOL) software (Fig. 2c, d). The potential distributions were calculated under open-circuit conditions. As shown in Fig. 2c, the potential difference between the charged skin and the Ecoflex surface of the TENG reaches a maximum when they are 30 mm apart. When the skin approaches the Ecoflex elastomer, the potential difference in the electrode decreases progressively. The change in the potential difference in the electrode impels free electrons to flow between the ground and the electrodes, thereby generating an alternating current signal[27]. The inclusion of an upper grounded shielding layer can effectively screen the electrical induction from the tribocharged skin, resulting in no obvious change in the potential difference in the electrode (Fig. 2d). The simulation results support the above description concerning the mechanism of electrostatic interference suppression by the carbon black-doped Ecoflex shielding layer.

To evaluate the screening effect of the shielding layer systematically, two modules with dimensions of $5 \times 5$ cm$^2$ were fabricated to separately represent the sensing node and elastic circuit of the UTE-skin. Typical output electrical signals of the two as-prepared samples were measured using an acrylic plate as the triboelectric positive material (Fig. 2e, f, and Supplementary Fig. 3a, b). An open-circuit voltage ($V_{oc}$) of 43 V and a short-circuit current ($I_{sc}$) of 280 nA were obtained when the acrylic plate touched the sensing node. Relatively high and stable voltage signals can be utilized as the sensing electrical signals of the UTE-skin. It was observed that $V_{oc}$ and $I_{sc}$ of the elastic circuit declined steeply to 8 V and 70 nA, respectively, upon inclusion of the top shielding layer, indicating the electrostatic screening effect of the shielding layer on the electrical signals. Furthermore, when the shielding layer was connected to the ground, the $V_{oc}$ was about 0.10 V, verifying that the electrostatic interference could be effectively screened by the grounded shielding layer. The misrecognition rate ($R_m$) in this study can be obtained by the formula $R_m = V_m/V_0$, where $V_0$ was the initial voltage value and $V_m$ was the voltage value of misrecognition signals. As shown in Supplementary Table 1, the proposed UTE-skin design delivered the lowest misrecognition rate of 0.20% than state-of-the-art TENG-based structures. Thus, the top shielding layer was validated as effective in eradicating the issue of misrecognition. These experimental findings are in good agreement with the simulation results shown in Fig. 2c, d.

Next, we systematically investigated the impact of carbon black content on the degree of electrostatic interference screening of the shielding layer. In Fig. 2g, h, we show the $V_{oc}$ and $I_{sc}$ of the elastic circuit modules at various carbon black mass fractions in the Ecoflex

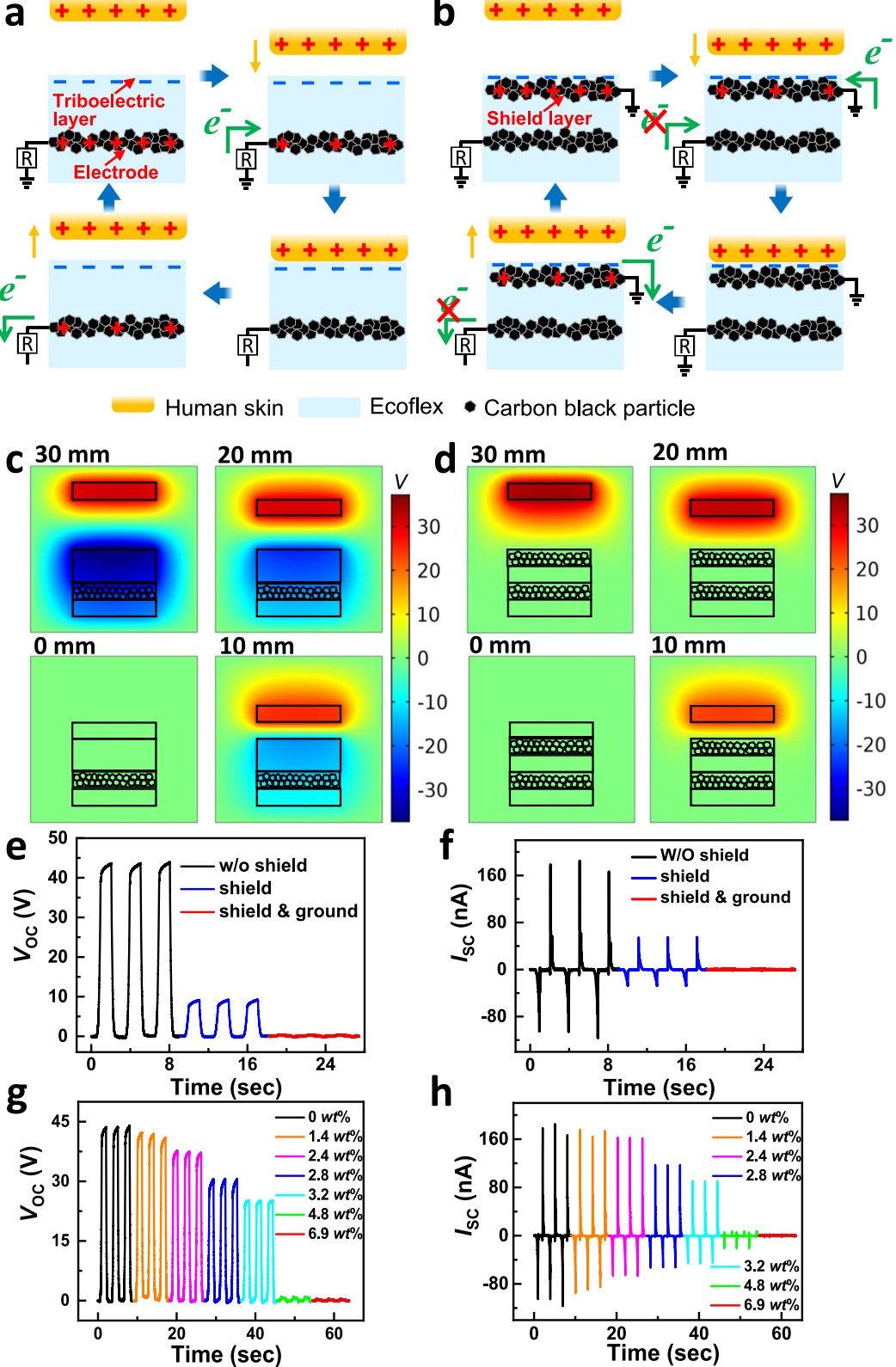

**Fig. 2 | Principle and electrical characterizations of misrecognition suppression by the shielding layer.** Schematic illustration of the working mechanism of UTE-skin operating at single-electrode mode (**a**) without and (**b**) with a shielding layer. Numerical calculations of the potential distributions at different distances between the two triboelectric surfaces (**c**) without and (**d**) with a shielding layer and calculated voltages *versus* distances. Comparison plots of the output (**e**) voltage and (**f**) current of the sensing nodes, elastic circuits with shielding layers, revealing the suppression of electrostatic interference by the shielding layer. Dependence of the output (**g**) voltage and (**h**) current of the elastic circuit on the mass loading fraction of carbon blacks in the shielding layer.

composites while maintaining identical experimental conditions. It reflects a sharp decrease in $V_{oc}$ and $I_{sc}$ values with increasing carbon black contents from $0\,wt\%$ (43 V and 280 nA) to $4.8\,wt\%$ (0.6 V and 20 nA), and then $6.9\,wt\%$ (0.3 V and 0.4 nA), as a result of the further reinforcement of the electrostatic shielding effect. As charge transport between carbon particles involves percolating particle-particle junctions[43,51], carbon black at $4.8\,wt\%$ content can provide more conducting pathways that facilitate the effective transmission of electrostatically inductive charges. When the carbon black content reached $6.9\,wt\%$, $V_{oc}$ and $I_{sc}$ decreased to approximately zero because all electrostatic induction charges of the circuits were screened by the highly conductive shielding layer. At this point, the misrecognition issue is less discernable, ensuring that the output signals are the exact signals generated by the sensing nodes. Thus, the accuracy and reliability of multiplexing UTE-skin can be significantly improved. These experimental results validate the reliability of the mechanism proposed in Fig. 2b. Furthermore, the effects of the thickness of the shielding layer on the electrical performance were investigated (Supplementary Fig. 4). Note that both $V_{oc}$ and $I_{sc}$ declined to near zero when the shielding layer of thickness ranging from ~0.2 to ~1.2 mm was grounded, indicating that the grounded shielding layer's capability of electrostatic interference screening was independent of its thickness. The thickness insensitivity of the shielding layer suggested the high practicality of the proposed electrical signal shielding strategy. Considering the dimension of the acrylic mold in the fabrication process of the UTE-skin, the thickness of the shielding layer adopted was ~1.0 mm.

## Mechanical characterizations of UTE-skins

Mechanical properties of self-powered triboelectric e-skins are critical for stretchable and untethered devices. As a first trial, to validate that the design can be utilized for deformable self-powered sensing arrays, the electrical performances of the above-mentioned as-prepared two modules were evaluated when undergoing uniaxial tensile strains from 0 to 100% (Fig. 3a–c and Supplementary Fig. 5). The $5 \times 5\,cm^2$ dimension of the as-prepared sensing node was selected to visualize the stretching deformation (Fig. 3a). We characterized the outputs triggered by applying a 0.4 KPa contact force driven by a custom-made motor and stretched the device while maintaining the same area of the impacting force. Figure 3b, c demonstrate the importance of the shielding layer for the performance of the UTE-skin. The applied uniaxial strain is defined as $\varepsilon_{applied} = \frac{L-L_0}{L_0}$, where $L_0$ and $L$ are the lengths of a unit module before and after the deformation, respectively. As the strain increased to 100%, the $V_{oc}$ of the sensing node slightly decreased from 43 V to 30 V, whereas $I_{sc}$ decreased from 130 nA to 40 nA (Fig. 3b, c). This is because, as the uniaxial strain increases, the elastic electrode deforms and its resistance increases in response to the cracks that developed at the particle-particle junctions, disrupting the conducting pathways and thus decreasing the net surface charges and the resulting electrical signals. Notably, with the shielding layer, the amplitude of the electrical output substantially diminishes when the elastic circuits are touched under the same tension (Fig. 3b, c). When the shielding layer was further grounded, $V_{oc}$ and $I_{sc}$ were nearly zero because all the triboelectric charges on the connected circuits were screened (Fig. 3b, c). In this case, the misrecognition of the stretched circuits remains to be eliminated. Consequently, the reliability and robustness of the UTE-skin can be considerably reinforced. In addition, the relatively large error bars existing in current signals in low-strain cases mainly resulted from the variations in contact time between external stimulus and Ecoflex triboelectric layer within different measurement times under identical stretching conditions (Supplementary Fig. 6). While the relatively stable voltage signals could be utilized as the sensing electrical signals of the UTE-skin in the self-powered sensing applications, which ensured the accuracy, robustness, and repeatability of the sensing platforms.

Subsequently, we examined the effect of the shielding layer when the modules were stretched biaxially. Figure 3d shows an optical image of the sensing node under a biaxial tensile of 100%, and the maintained mechanical integrality reveals extraordinary extensibility. The modules also exhibited outstanding mechanical endurance when undergoing 100% biaxial stretching, with the $V_{oc}$ and $I_{sc}$ remaining notable values of 28 V and 40 nA, respectively (Fig. 3e, f), which can be utilized as the sensing signals. Similarly, when the elastic circuits withstand the same biaxial tensile stress, $V_{oc}$ and $I_{sc}$ are approximately zero, indicating the excellent deformability and durability of the shielding layer. This verifies once again that the stretchable shielding layer can enhance the stability, accuracy, and repeatability of UTE-skin, enabling it to work stably under biaxial tensile strains of up to 100%.

Furthermore, self-powered e-skins can be stretched in all directions, which is a key feature for untethered practical applications. This feature ensures that the detection of uniaxial or biaxial strains is not affected by unexpected strains. A customized setup was designed to examine the isotropic strain with dynamic inflation driven by the air pressure (Fig. 3g). The isotropic strain, $\varepsilon_A$, is defined as $\varepsilon_A = \Delta A/A_0$, where $\Delta A$ is the area variation and $A_0$ is the original area[52]. Because the dynamic inflation was actuated by air pressure and the thin shielding layer was freestanding before inflation, it was difficult to identify the isolation strain when the radius of curvature, $\rho$, was 0 cm. Thus, an isotropic strain when $\rho = 5$ cm was selected as the basic strain. When $\varepsilon_A$ increased to 400%, the $V_{oc}$ and $I_{sc}$ of the sensing node still achieved high values of 24 V and 40 nA, respectively (Fig. 3h, i), indicating that the UTE-skin can retain its sensing functionality even when subjected to high isolation strains of up to 400%. Furthermore, the electrical signals of the elastic circuit with a grounded shielding layer were nearly zero even when the isotropic strain to 400% (Fig. 3h, i and Supplementary Fig. 5c). Taken together, the above results reveal that the UTE-skin performs excellently under isotropic strain. This ability enables self-powered e-skins to sense external stimuli independent of touching the sensor platforms directly.

Considering the pivotal function of the conductive shielding layer in eliminating misrecognition, the effects of carbon black content on the resistance of the shielding layer were further investigated. Figure 3j shows the resistance to various carbon black mass fractions at varied strains from 0% to 100%. The resistance of the shielding layer changed slightly when the carbon black content increased from $1.4\,wt\%$ to $3.2\,wt\%$ ($\sim 10^{11}\,\Omega$). As the carbon black content further increased, the average resistance decreased by four ($\sim 10^7\,\Omega$, $4.8\,wt\%$) and five ($\sim 10^6\,\Omega$, $6.9\,wt\%$) orders of magnitude. These enhancements in conductivity correlate with the degree of carbon particle percolation, as shown in Supplementary Fig. 7. The resistance of the shielding layers with carbon black contents $< 4.8\,wt\%$ was barely affected by the external strain, owing to the insulating properties ($\sim 10^{11}\,\Omega$). In addition, normalized resistance changes ($\Delta R/R_0$) in the shielding layers with carbon black contents of $4.8\,wt\%$ and $6.9\,wt\%$ at varying uniaxial strains are also shown in Fig. 3k. Here, the normalized resistance change is defined as $\Delta R/R_0 = (R - R_0)/R_0$, where $R_0$ and $R$ are the resistances of the shielding layer before and after the deformation, respectively. The shielding layer with $4.8\,wt\%$ carbon black with $R_0$ of $10^7\,\Omega$ had an abrupt resistance change, with $R/R_0$ being 3.5 at 10% strain, 16.0 at 50% strain, and 441.0 at 100% strain, respectively. However, at $6.9\,wt\%$ carbon black contents with $R_0$ of $10^6\,\Omega$, the $R/R_0$ was 1.6 at 10% strain, 2.0 at 50% strain, and 2.6 at 100% strain, respectively. We anticipate that at $4.8\,wt\%$ carbon black, fractures that developed at particle-particle junctions deteriorate the conducting pathways, which could be a source of sudden resistance changes upon deformation[43]. As the mass fraction increased to $6.9\,wt\%$, there was a smaller degradation in conductivity because the excessive carbon particles significantly strengthened the cross-linking, and the formation of fractures was dampened owing to the stiffening effect of the particles[51]. Thus, we can conclude that the mass fraction of carbon

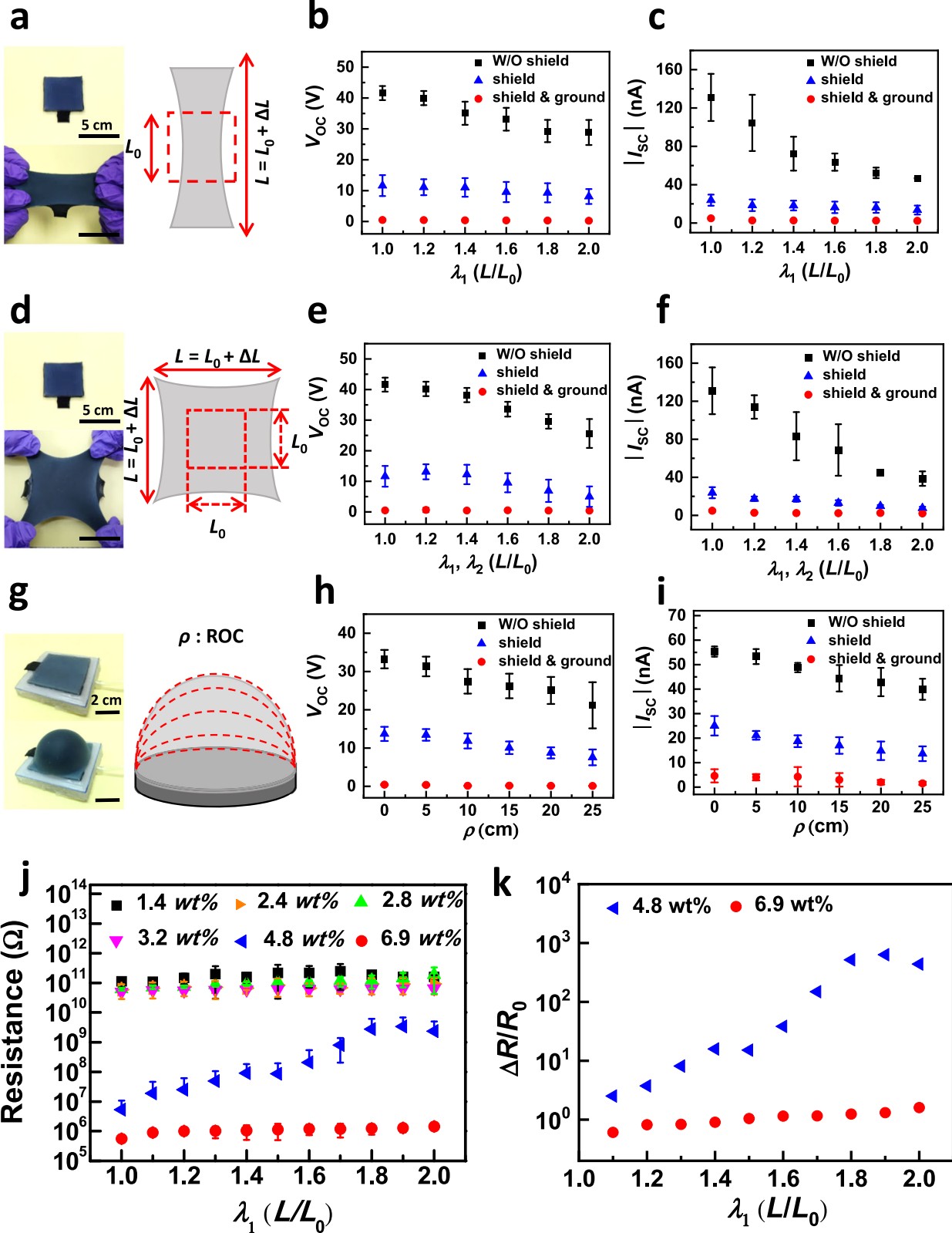

**Fig. 3 | Electrical performance characterization under uniaxial, biaxial, and isotropic tensile strains. a** Optical photograph (left) and schematic diagram (right) of the sensing node module (5 × 5 cm²) after a uniaxial strain of 100%. **b** Voltage and (**c**) current signals of the sensing nodes, elastic circuits, and grounded elastic circuits as a function of uniaxial strain. **d** Optical photograph (left) and schematic diagram (right) of the sensing node module (5 × 5 cm²) after a biaxial strain of 100%. **e** Voltage and (**f**) current signals of the sensing nodes, elastic circuits, and grounded elastic circuits as a function of biaxial strain. **g** Optical photograph (left) and schematic diagram (right) of the sensing node module (5 × 5 cm²) after isotropic strain. **h** Voltage and (**i**) current signals of the sensing nodes, elastic circuits, and grounded elastic circuits as a function of isotropic strain. **j** Resistance-strain characteristics of the carbon-blacks-doped Ecoflex shielding layer with different carbon-black contents. Error bars in (**b**, **c**, **e**, **f**, **h**, **i**, **j**) represent standard deviations, $n = 5$ independent samples. **k** Dependence of normalized resistance changes of the shielding layer with 4.8 and 6.9 $wt\%$ carbon blacks on various strains up to 100%.

black in the Ecoflex composite determines the conductivity, particularly when strain is applied externally. Furthermore, the shielding layer requires high conductivity (6.9 $wt\%$ carbon black) to efficiently transport electrostatic charges and eliminate misrecognition interference among adjacent elements and circuits. These findings agree well with the experimental data summarized in Fig. 2g, h.

To further assess the durability of the design for UTE-skin, the electrical performance (when subjected to a mechanical force of 0.4 KPa) of the sensing node module was evaluated after 1000 cycles of uniaxial stretching up to 100% strain (Supplementary Fig. 8). The sensing node can operate stably in the initial and stretched states because the output voltage shows no apparent degradation after repeated contact-separation motions, demonstrating its exceptional robustness. The resistance of the shielding layer (6.9 $wt\%$ carbon blacks) varied only slightly when stretched to 100% strain after 1000 repeated cycles (Supplementary Fig. 9). Cyclic durability is sufficient for applications in which sustaining only a few gross deformations is of paramount importance.

In practical applications, the UTE-skin can undergo mechanical contact with various materials. Therefore, the electrical outputs of the sensing nodes with different contact materials were evaluated, including Kapton, acrylic, glass, aluminum (Al), paper, copper, FEP, and textiles (Supplementary Fig. 10a, b). Kapton exhibits more tribopositive properties compared to acrylic, glass, aluminum (Al), paper, copper (Cu), FEP, and textile as it loses electrons by contact electrification, and it shows a positive triboelectric voltage. It should be noted that stable and substantial electrical outputs can be generated by the UTE-skin when contacting metallic materials, such as Al (29 V, 107 nA) and Cu (25 V, 97 nA). The active electrical signals by UTE-skin in response to stimuli from metallic materials supported the feasibility of a robotic skin, as depicted in Fig. 1c.

## Stretchable sensor networks for pressure mapping and touch recognition

Self-powered triboelectric tactile sensor arrays based on the single-electrode mode are promising for obtaining a high spatial resolution for touch imaging and pressure mapping. However, misrecognition arising from the electrostatic induction of high-density electrode lines remains a pivotal obstacle that drastically restricts their actual usage for accurate position recognition in response to contacting objects.

To investigate the capability of the UTE-skin as a self-powered mechanical signal sensor, we studied its output behavior depending on various operating frequencies and applied pressures. As indicated in Supplementary Fig. S11a, $V_{oc}$ displayed a steady output of 43 V with increasing frequencies from 0.25 to 4 Hz. While the short contacting time (~2 s) at high frequency resulted in a fast charge flow, responsible for the elevated $I_{sc}$ from 0.28 to 1.70 μA (Supplementary Fig. S11b). Supplementary Fig. S12 depicted the $V_{oc}$ under 0.25 Hz with applied pressures of 0.2, 0.4, 0.8, and 1.2 KPa, which yielded 25.5, 45.0, 53.6, and 64.7 V, respectively. The relevant $I_{sc}$ attained 0.50, 1.40, 1.80, and 2.02 μA, validating the electrical outputs depended on the applied pressures. Meanwhile, as shown in Supplementary Fig. S13, $V_{oc}$ and $I_{sc}$ increased linearly as the applied pressures increased from 0.2 to 2.0 KPa, whereas the growth rate slowed with the applied pressure augmenting to 3.2 KPa. With such high electrical performance, the UTE-skin was able to instantaneously power up over 40 LEDs in series by gentle touching (Supplementary Fig. 14 and Movie 1). In addition, the gathered mechanical energy by UTE-skin could also drive an electronic watch after charging for only ~200 s (Supplementary Fig. 15). These results confirmed that the UTE-skin can be used as an energy harvester, serving as an efficient power supply that satisfies differential operation requests from users. Meanwhile, it can produce notable and robust electrical signals even under slight pressures with low frequencies, ensuring its feasibility as a stretchable sensor network for self-powered mechanical signal sensing.

To demonstrate high-resolution tactile sensing imaging, a large-area matrix of 4 × 4 pixelated UTE-skin with an area of 25 × 25 cm² was fabricated to achieve full-range pressure monitoring based on the voltage and color of each pixel (Fig. 4a, b). Experimentally, different applied pressures have been exerted by using different sizes of contact letters "N," "C," "H," and "U" on the surface of the sensing array with and without a top shielding layer. The detected sensing signals can provide information regarding the positions, and the peak of the electrical response curve of each unit matches the local pressure response at that coordinate. However, the sensor array without a shielding layer cannot locate the contact letters because of excessive misrecognition signals from the high-density connecting wires (Fig. 4a). Using the carbon black-doped Ecoflex elastomer as the upper shielding layer can effectively prevent misrecognition interference (Fig. 4b). The letters can be clearly identified between the touched and untouched pixels with reliable electrical output from the sensing array. The shielding layer on the non-sensing areas greatly eliminated the misrecognition of the electrode lines, leading to a substantial improvement. These results validate that the carbon black-doped Ecoflex shielding layer works well for eliminating misrecognition, which exhibits pronounced advantages for localizing external stimuli with high spatial resolution.

Further, experiments on recognizing the letter "N" by the UTE-skin when writing in real-time with a finger were performed. As shown in Fig. 4c–e, when writing on the UTE-skin without a shielding layer, the electrode circuits generated noticeable interfering signals, which conveyed incorrect and misleading information about the physical stimulus, severely affecting recognition fidelity. However, with a top shielding layer, the movement trajectory of the finger can be clearly recognized without interference from neighboring circuits according to voltage signals from different sensing pixels in a time-sequential manner (Fig. 4f–h). This confirms that UTE-skin with a shielding layer can accurately recognize the motion trajectories of finger sliding, which offers great potential for self-powered real-time haptic sensing of external stimuli.

To enhance the integrability of sensory systems, the device resolution was further investigated by programming sensing node spacing from 0 to 5 mm (Fig. 4i, j). Notably, as the electrode spacing gradually increased to 3 mm, the anti-interference index approached 96.0% (Fig. 4k), demonstrating the high resolution of the UTE-skin sensor array. It enabled large-area, conformal precise tactile recognition and pressure mapping applications of the UTE-skin. Moreover, the extracted variation in sensing resolution versus spacing was believed to guide the operational effectiveness of the UTE-skin for use at the system level.

## Self-powered active sensing of stretchable triboelectric e-skin

To demonstrate the practical feasibility of the elastic e-skin, we fabricated a stretchable UTE-skin-based smart glove. Figure 5a shows a structure comprising five individual sensing nodes (2 × 2 cm²) and elastic conducting wiring (width of 1 cm) laminated on a stretchable textile substrate. Thus, the sensing elements can be segregated from the strain when the entire system is stretched. The sensor sheet measured 40 × 10 cm² and could be readily placed on a person or robot by wearing it. The output of the sensor array was transmitted through the elastic electrodes. As indicated in Fig. 5a, touching the conducting wires without shielding also produces pronounced electrical signals, which might be erroneously identified as actual signals induced by the sensing units. Figure 5b shows a glove sensor system with a top shielding layer that exhibits excellent stretchability and mechanical durability. Importantly the electrical signal is approximately zero when the circuits are touched upon the shielding layer, signifying that the misrecognition interference is greatly eliminated. That is, the output signals accurately detected by the sensor system are those generated from the interactions of the human skin with the sensing nodes, rather than any other undesired signals from the connected circuits. Overall,

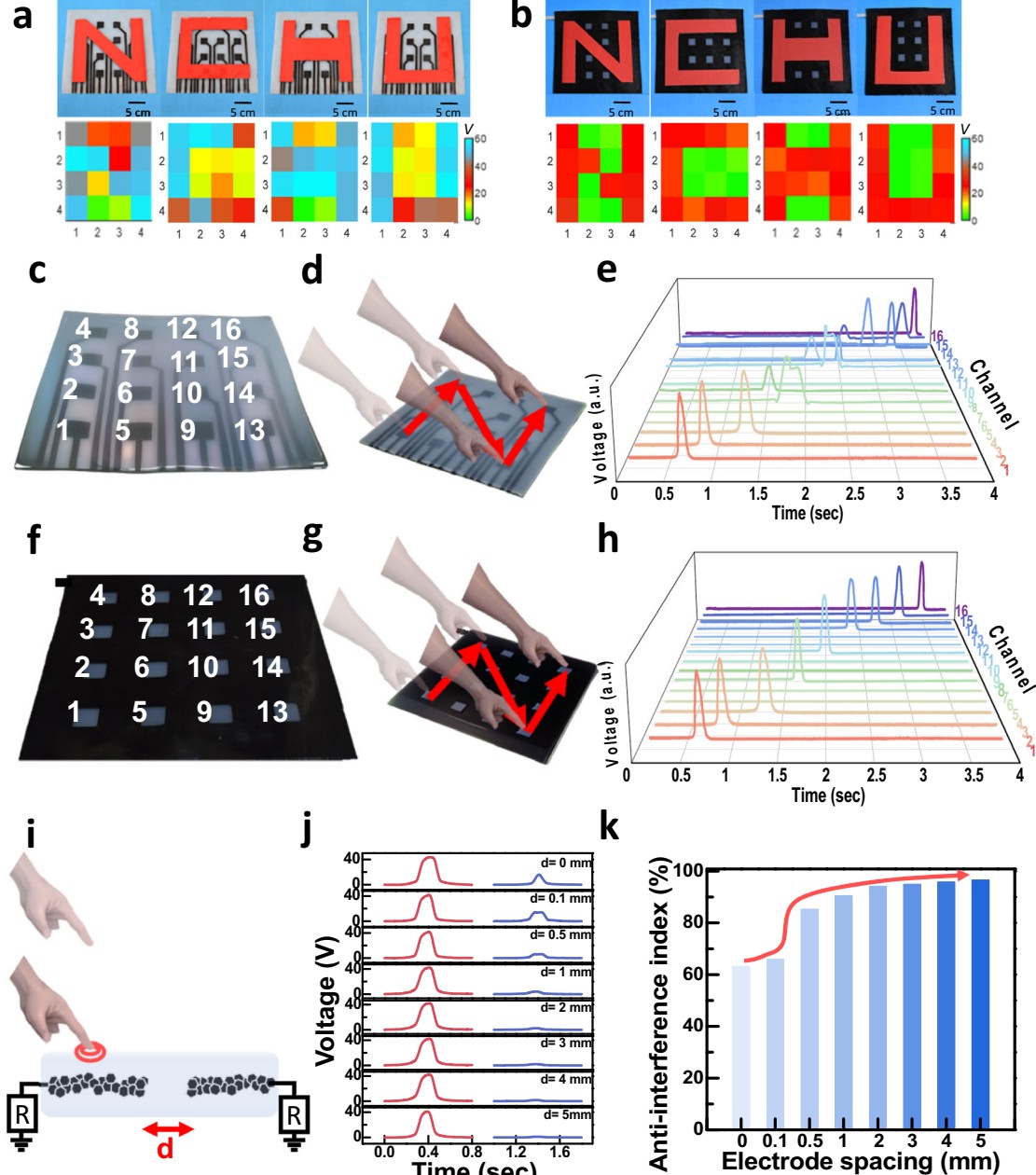

**Fig. 4 | Stretchable sensor networks for pressure mapping and touch recognition.** Schematic diagram (upper) and top views of 2D mapping for output voltage intensity (bottom) response to different applied pressure exerted using different sizes of contact letters "N", "C", "H", and "U" of the surface of 4 × 4 pixelated UTE-skin (**a**) without and (**b**) with a top shielding layer. **c** Photograph of a large-area matrix of 4 × 4 pixelated UTE-skin without a shielding layer (25 × 25 cm²). **d** Schematic of sliding mode and (**e**) output voltages of pixels in the moving trajectory of the finger writing the letter "N" on the surface of UTE-skin without a shielding layer. **f** Photograph of a 4 × 4 pixelated UTE-skin with a shielding layer (25 × 25 cm²). **g** Schematic of sliding mode and (**h**) output voltages of pixels in the moving trajectory of the finger writing the letter "N" on the surface of UTE-skin with a shielding layer. The term (a.u.) in (**e**, **h**) represents the arbitrary unit. **i** Schematic diagram of the electrical signal tests of UTE-skin with two adjacent sensing panels spaced $d$ apart. **j** Output voltage signals of the two adjacent sensing nodes with different electrode spacing by touching the left one. For a fair comparison, the tapping sites highlighted at the left panel were operated with a uniform contact pressure and area. **k** Plots of the statistical anti-interference index of UTE-skin sensory system relying on different spacings from **j**, displaying the anti-interference abilities of tactile sensors at each spacing to improve the system-level design. Note that the index was extracted from the ratio of $(V_O - V)/V_O$, where $V$ and $V_O$ were the measured voltage values for each cell with possible disturbance and without disturbance, respectively. A curved arrow was utilized to guide the eyes.

the outstanding anti-interference properties of the shielding layer provide effective isolation from the sensing nodes and circuits, ensuring low noise levels in the self-powered tactile sensing system.

In Fig. 5c, four sensing nodes (2 × 2 cm²) were fabricated and arranged on the back and front sides of a smart insole to reflect the pressure distribution, dynamic motion, and mapping in real-time to improve the intelligence of gait analysis. We first monitored and

recorded the trajectories of moving objects to demonstrate the intriguing features of this smart insole. When a 500 g weight wood cylinder was rolled across the insole (Fig. 5d), the pressure changed, and the trajectory profile was recorded in real-time. Apparently, without a shielding layer, the motion path of the weight, moving from the heel (mark No. 1 in Fig. 5e) to the lateral (the largest dashed circle), until the forefoot (marks No. 2, 3, and 4), is far from distinguished in the graph

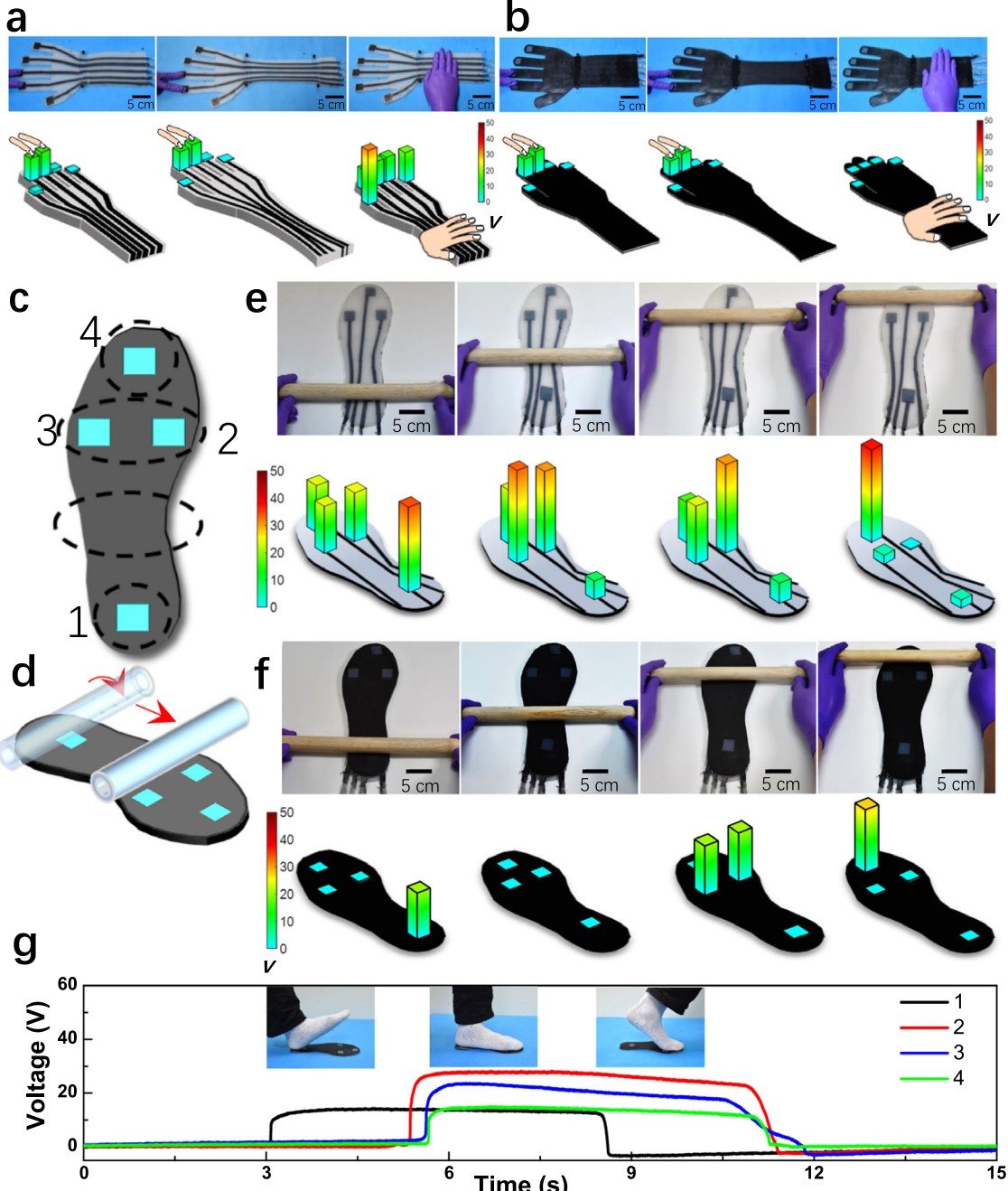

**Fig. 5 | Stretchable sensor networks for stretchable robotics.** Photographs (upper) and 2D voltage mapping (bottom) of the unstretched and stretched smart glove sensor networks (**a**) without and (**b**) with a top shielding layer in response to human finger touch. **c** Schematic diagram of the position of selected four sensor nodes on a stretchable insole. **d** Schematic of a cylinder rolling along different pixels. Optical images (upper) and voltage mapping (bottom) of the selected four sensor nodes (**e**) without and (**f**) with a top shielding layer in the moving trajectory of the cylinder. **g** Real-time measurement of the selected four sensor nodes under working motion, the inset parts display the different gait phases.

owing to the large misrecognition interference (Fig. 5e). Notably, as shown in Fig. 5f, the misrecognition issue of the insole was solved using the top shielding layer. The change in dynamic charges and the corresponding output voltage can be monitored in real-time with high resolution using a large-area sensor array. Hence, the shielding layer in our intelligent insole performs a crucial function in promoting signal accuracy by minimizing undesirable misrecognition signals.

As shown in Fig. 5g, the three insets illustrate the photographs of three walking states represented by "heel striking," "standing phase," and "pushing off through toes" during a single walking cycle. According to the working principle of TENGs, all the sensing nodes on

the insole are incessantly pinched and released to generate triboelectric signals. The response from the UTE-skin at point 1 (black line) is distinctly ahead of the other points, as the heel touches the ground first during walking. Thereafter, three voltage signals were detected from the other three nodes (red, blue, and green lines) associated with the standing contact and heel-off states, which revealed the magnitude of the pressure exerted by the volunteer as he walked forward. Notably, all electrical signals in Fig. 5g closely mirror the walking patterns delineated above. Collectively, these demonstrations highlight the vast potential of our shielding layer incorporating triboelectric e-skin-based insoles for smart healthcare applications with high accuracy on a

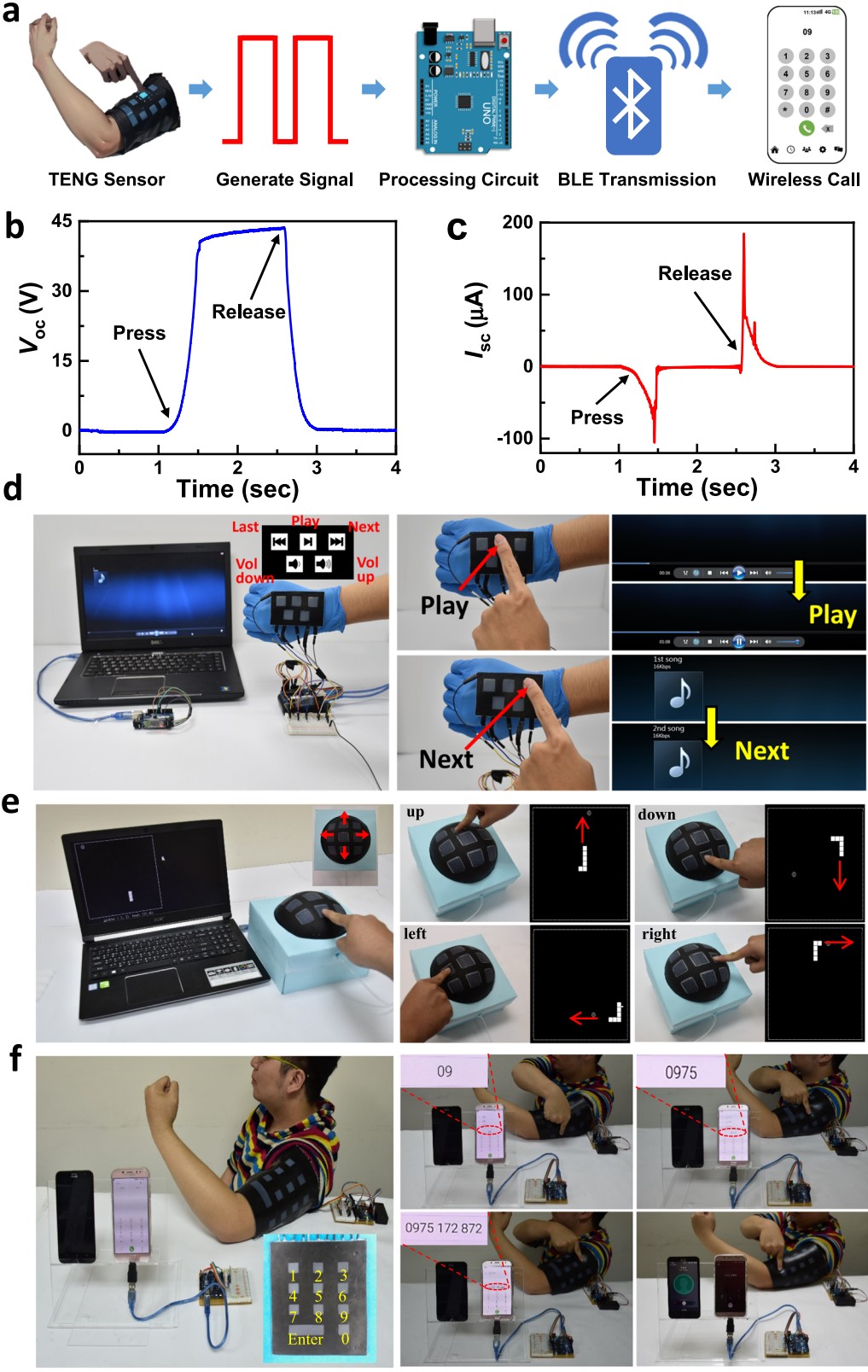

**Fig. 6 | System-level applications of the UTE-skin self-powered sensor arrays.** **a** Schematic diagram of the actively signal progressing system of the UTE-skin for human-machine interfaces. **b** Output voltage and (**c**) current of the UTE-skin upon finger touching and releasing. **d** Demonstration of operating the "Play" and "Next" instructions via triggering the self-powered UTE-skin. **e** Photograph of a spheric game controller with a 4 × 4 pixelated array and various demonstrations of the UTE-skin sensor array-enabled operating instructions, including "up", "down", "left", and "right". **f** Optical image and demonstrations of the UTE-skin sensor array using a self-powered active wearable cell phone panel.

large scale, particularly in footwear design, exercise/sport ballistic data pickup, injury prophylaxis, and diabetic ulcer prediction.

**Soft haptic interfaces for virtual and augmented reality**

The excellent deformability and conformability of the large-area stretchable multiplexing triboelectric e-skin with a deformable shielding layer enable its application in cost-effective haptic devices for VR/AR use. As depicted in Fig. 6a, in the system-level human-machine applications of the UTE-skin, the active electrical signals were produced by the TENG sensing node upon external stimuli, and the signal processing circuit and wireless signal transmitter were driven by an external power source, enabling multifunctional sensing functions of the system. Figure 6b, c shows the electrical signals generated when a human finger touches and releases a UTE-skin-sensing node.

A wireless UTE-skin-based music controller attached to the back of a human hand was further demonstrated. Five multifunctional sensing nodes ($1.5 \times 1.5$ cm$^2$) were used as the "Play", "Next", "Last", "Volume up", and "Volume down" keys, respectively. Lightly touching the second or the third sensors freely controlled the instructions of "Play" or "Next" (Fig. 6d and Supplementary Movie 2). In Fig. 6e, we also demonstrated a soft spherical haptic interface that can automatically control a video game (termed "Snake Game") in real-time, where the integrated system comprises nine arrow keys fabricated using the TENG sensors. The untethered soft interface combined the optimal traits of physical buttons. A pressurized elastomeric membrane enabled tactile interaction while delivering touch-sensing functionality. In coordination with a microcontroller, pressing each operating key initiated the launch of the TENG sensor and therefore executed the corresponding commands, like "up," "down," "left," and "right" (Supplementary Movie 3). No noticeable misrecognition interference was observed, and the movement of the snakes coincided closely with the sweeping motion of the human finger.

A wearable smartphone keypad on a garment was further demonstrated as a wireless self-powered controller by integrating 11 TENG tactile sensors (10 of which were employed as numeric keys and the other as the dial key) (Fig. 6f). After dialing several real phone numbers, the mobile phone was successfully accessed (Supplementary Movie 4). Such demonstrations of self-powered microcontrollers suggest promising uses for our shielding layer incorporating a multiplexing triboelectric e-skin in the field of interactive haptic feedback control, supporting further research geared toward intelligent robotics.

## Discussion

In summary, we developed an effective electrical screening strategy complementing eliminating e-skin misrecognition between sensing nodes and internal connecting wiring. A large-scale, untethered, and intrinsically stretchable multiplexing triboelectric e-skin ($4 \times 4$ pixels in a 25 cm $\times$ 25 cm area) was constructed by seamlessly integrating all compliant material ingredients via a scalable and facile manufacturing process, targeting precise tactile imaging and pressure mapping. The entire triboelectric e-skin with a single-electrode mode comprised elastomeric Ecoflex as the triboelectric layer and matrix and elastic carbon black-doped Ecoflex as the top shielding layer, electrodes, and circuits. Specifically, with the design of a conductive nanocomposite-based omnidirectionally stretchable and deformable shielding layer, the UTE-skin realized an ultralow misrecognition rate of 0.20%. Moreover, it retained its functionality even under 100% uniaxial, 100% biaxial, and 400% isotropic strains as well as achieved clear pressure imaging and real-time visualization of multiple-point touch. Experimental and numerical studies have proven that the notable anti-interference ability originates from the effective suppression of misrecognition signals of the electrode lines by the shielding film. This efficiently provides isolation among the pixels and circuits and thus elevates the overall accuracy, reliability, and

repeatability. Demonstrations of the multifunctional e-skin in smart wearable gloves to identify contacting objects, and intelligent insoles to monitor and analyze human gait illustrate its robust capabilities. In parallel, it enables the scalable construction of system-level sensing arrays toward human-machine interactions, including a compact music controller, a soft spherical real-time game controller, and a wearable smartphone keypad. This unique device design surmounts the limitations of misrecognition interference and modulus mismatch of each functional layer, representing a tremendous leap that will present an enticing avenue for the scaled-up fabrication of large-area stretchable multiplexing e-skins capable of diverse functionalities.

## Methods

### Ethics declaration

All human subject studies were approved by the National Chung Hsing University and the volunteers gave informed consent. The authors affirm that human research participants provided informed consent for publication of the images in Fig. 6f.

**Fabrication of UTE-skin.** Liquid silicone rubber was obtained by mixing the silicone base and curing it in a volume ratio of 1:1 (Ecoflex super-soft silicone 0030, Smooth-On, Inc., PA, USA) in a beaker. Carbon blacks were then added to the beaker and stirred to obtain a uniform conductive silicone. The volume ratio of carbon blacks to Ecoflex was 1:1 (weight ratio: 1:13.4). The carbon black-doped Ecoflex shielding layer was smeared from its solution over a flat acrylic mold at 30 °C for 6 h. After complete curing, the shielding layer was determined and cut into a $4 \times 4$ sensing array. The pristine liquid Ecoflex was poured onto the patterned shielding layer as the triboelectric layer. Another acrylic mold was tailored into a $4 \times 4$ array, and the corresponding interconnects were laminated onto the as-prepared composite in parallel. Carbon blacks-doped Ecoflex was poured into its groove to construct the elastic electrodes (*i.e.*, the sensing nodes) and connecting circuits. After casting a layer of Ecoflex elastomer as the substrate, peeling off the entire device from the acrylic mold, and turning it upside down, untethered UTE-skin was ultimately obtained.

**Characterization.** The mechanical properties were examined using an Instron 5943 instrument. Resistance was measured using a Keithley 2400 source meter. The open-circuit voltage ($V_{oc}$) and transferred charge ($Q_{tr}$) were measured using a Keithley 6514 electrometer and the short-circuit current ($I_{sc}$) was measured using a Stanford low-noise current amplifier (model SR570). For the standard measurement of the electrical output, contact materials were used to touch the UTE-skin using a commercial linear mechanical motor at a constant contact force. The applied force was measured by using a Vernier LabQuest Mini instrument. The contact materials were initially placed 30 mm away from the device, at which point the potential difference was zero. To demonstrate the human-machine interfaces, the UTE-skins were integrated with Arduino Uno microcontrollers to process the generated electrical signals.

### Reporting summary

Further information on research design is available in the Nature Portfolio Reporting Summary linked to this article.

## Data availability

All data are available in the main text or the supplementary materials. Source data are provided with this paper. Any additional requests for information can be directed to and will be fulfilled by the corresponding authors. Source data are provided with this paper.

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

## Acknowledgements

B.S. and M.-H.L. contributed equally to this work. The authors thank the experiment assistance by H.-J.L. and H.-M.W.; This work is financially supported by National Science and Technology Council (111-2221-E-005-060; 112-2811-M-005-016; 112-2221-E-005-038), "Innovative Center on Sustainable Negative-Carbon Resources" from The Featured Areas Research Center Program within the framework of the Higher Education Sprout Project by the Ministry of Education (MOE) in Taiwan, and National Natural Science Foundation of China (52103306), Natural Science Foundation of Jiangsu Province (BK20210719), China Postdoctoral Science Foundation (2023M732534) and Postdoctoral Excellence Programme in Jiangsu Province (2023ZB510).

## Author contributions

Y.-C.L. and B.B.S. conceived and designed the experiments and led the research. M.-H. L. contributed to the device fabrication and electrical measurements. B.Q.S., R.Y.L., and Y.-C.L. supervised the research. T.-C.W., W.-C.P., T.-Y.K., Y.-C.H., and J.-Y.C conducted the experimental work. B.B.S., M.-H. L. and Y.-C.L. analyzed the data. B.B.S., M.-H. L. B.Q.S., R.Y.L., and Y.-C.L. prepared and revised the manuscript. All the authors discussed the results and commented on the paper.

## Competing interests

The authors declare no competing interests.
