## [Peer Review File · Nature Communications]

REVIEWER COMMENTS

Reviewer #1 (Remarks to the Author):

This article utilizes carbon black-doped conductive Ecoflex and pure Ecoflex, as well as patterned grounded carbon black-doped conductive Ecoflex layers, to distinguish friction-generated stimuli from the signal by non-electrified regions. This method enables the realization of a high-resolution tactile pressure sensing array based solely on intrinsically stretchable materials. The authors also demonstrate the potential applications of stretchable sensors and human-machine interfaces based on the foundation of the reported technology. Overall, I feel this work is well prepared and organized. There are several issues need to be addressed before publication:

1. Self-powering refers to the utilization of friction-generated electricity to collect energy and supply it to microcontrol systems (Adv. Funct. Mater. 2021, 31, 2100709, Adv. Mater. 2022, 34, 2200724.). While in this article, the collection of triboelectric signals are achieved using an external power source (such as the system-level applications demonstrated in Figure 5), So the system reported in this work is not a fully self-powering system?

2. Crosstalk refers to the phenomenon where a signal being transmitted through one wire can affect nearby wires, leading to unexpected changes in the signals of those wires.

a) The first type of crosstalk mainly occurs in a positive-negative cross-electrode array. When current flows through a sensor, it can pass through other sensors due to the absence of barriers, leading to crosstalk in the array of sensors. This paper uses a single-electrode model with a common ground, and there is no positive-negative electrode crossover. Therefore, this type of crosstalk does not exist.

b) The second type of crosstalk occurs when there is a close spacing between two wires, leading to the presence of parasitic capacitance or coupled inductance. When a signal is transmitted through one wire, it can cause signal changes in the other wire due to the interaction between them. The paper indicates that the spacing between electrodes and wires is relatively large, with analysis shown in the diagrams to be around 1 cm or more. Therefore, this type of crosstalk also does not exist in this case.

According to the paper, the signal interference mentioned primarily arises when there is no insulation provided by conductive carbon-black doping Ecoflex. In this case, the wires also act as electrodes and can generate frictional electric signals. Therefore, similar phenomena as shown in Figure 4e can occur. This type of interference may not be that significant in other designs, please comment on that.

3. According to Figure 1b, the "elastic triboelectric layer" and "triboelectric layer" seem to be considered as a single entity. The substrate is divided into two parts, and only the negative part of the triboelectric effect is mentioned, which is quite confusing.

4. The paper mentions that Figures 1e and 1f exhibit numerous efficient conducting pathways. However, without additional information or access to the complete paper, it is challenging to provide specific details on how these conducting pathways are achieved.

5. Figures 2a and 2b are not clearly annotated, making it difficult to understand their content easily.

6. In Section 2.2, the human skin is considered as a positively charged layer. Therefore, the presence of the robot in Figure 1c can be quite misleading.
7. Figures 3b and 3c display the potential and current information of three triboelectric sensors.
8. Figure 3k demonstrates the good electrical properties of conductive Ecoflex. However, there is a problem with understanding this. Even if we assume that the conductivity remains constant during the stretching process, the cross-sectional area of the material would decrease as it is stretched. According to the principles of incompressible materials, the cross-sectional area would shrink to at least 50% of its original size, resulting in an increase in resistance by at least twice or more. Additionally, why does the initial resistance start below 1?
9. In Figures 3c and f, larger error bars exist in current signals in low-strain cases. Why? Will it affect the sensing result in practical applications?
10. In Figure 4, the UTE-skin is applied as a stretchable sensor network for mechanical signal sensing. The study of the response of stretchable sensor networks to different mechanical signal inputs is indeed crucial. The authors should include quantitative experiments and discussions on this part.

Reviewer #2 (Remarks to the Author):

This manuscript reports a large-scale triboelectric nanogenerator (TENGs) tactile sensing array, named untethered triboelectric electronic skin (UTE-skin). This UTE-skin, featuring a 4x4 pixel array over a 25 cm x 25 cm area, combines self-powered sensing with exceptional stretchability, durability, and precise spatial accuracy. Made from flexible electronic materials like Ecoflex and carbon black-doped Ecoflex, it effectively minimizes crosstalk between sensors, ensuring stable and accurate readings even under extreme tensile strains. Applications include intelligent gloves for gesture recognition, smart insoles for gait analysis, and immersive training tools for human-machine interfaces. However, the reviewer recommends that this manuscript can be accepted in 'Nature Communications' with the following Minor Revision.

1. While the presence of the carbon black-doped Ecoflex layer has been confirmed to improve misrecognition rates, additional explanation is needed to show how it compares with other TENG-based structures in terms of reduced misrecognition. Please demonstrate that the recognition rate of the UTE-skin is superior under the same conditions used in experiments reported in other studies.
2. In Figure 4a, the author has implemented the recognition of contact letters with the UTE-skin. It would be more appropriate to demonstrate the usability of e-skin by showing how accurately it can recognize letters when writing in real-time with a finger.
3. The successful implementation of a large-area UTE-skin is well demonstrated. However, to further showcase its high applicability as an e-skin, it's necessary to develop and test a smaller version of the UTE-skin. For example, although the demo in Figure 5e is conducted on a large area, data is needed for the creation and implementation of an e-skin small enough to be attached to the back of a hand. Additionally, please provide recognition rates for different distances between pixels to demonstrate the resolution of the e-skin.

4. Since the title refers to 'Self-powered triboelectric skin', it would be beneficial to add data showing the extent of the UTE-skin's performance as an energy harvester.

5. Several references can be added to help the readers to understand this manuscript.

- Science and technology of advanced materials 20 (2019), 758-773

- Nano Energy 95 (2022) 107056

- Nano Energy 56, 531-546 (2019)

- Adv. Funct. Mater. 2022, 2112241

Reviewers' Comment Letter

Responses to Reviewer #1:

Reviewer #1: This article utilizes carbon black-doped conductive Ecoflex and pure Ecoflex, as well as patterned grounded carbon black-doped conductive Ecoflex layers, to distinguish friction-generated stimuli from the signal by non-electrified regions. This method enables the realization of a high-resolution tactile pressure sensing array based solely on intrinsically stretchable materials. The authors also demonstrate the potential applications of stretchable sensors and human-machine interfaces based on the foundation of the reported technology. Overall, I feel this work is well prepared and organized. There are several issues need to be addressed before publication.

Response: We sincerely thank Reviewer #1 for carefully reviewing our manuscript and raising professional comments that are significant to further enhance our work and improve the quality of the manuscript.

Comment 1:

Self-powering refers to the utilization of friction-generated electricity to collect energy and supply it to micro control systems (Adv. Funct. Mater. 2021, 31, 2100709, Adv. Mater. 2022, 34, 2200724.). While in this article, the collection of triboelectric signals are achieved using an external power source (such as the system-level applications demonstrated in Fig. 5), So the system reported in this work is not a fully self-powering system?

Response 1:

The authors thank Reviewer #1 for the comments regarding the system-level application demonstrated in original Fig. 5 (revised as **Fig. 6**), and we agree that it is not a fully self-powered system. As mentioned by Reviewer #1, a fully self-powered sensing system refers to a sensor being powered by the harvested energy (e.g. friction-generated electricity) [Adv. Funct. Mater. 2021, 31, 2100709; Adv. Mater. 2022, 34, 2200724]. In addition, self-powering sensing also represents that the sensor is activated directly by external stimuli without an electrical power supply [Matter 2021, 4, 116; Nat. Rev. Mater. 2022, 7, 870]. TENGs rely on the triboelectric effect deriving from a conjunction of contact electrification and electrostatic induction. When two materials come in contact, the TENG can transform mechanical energy into electrical signals with a relatively high amplitude, enabling its self-powered active tactile recognition and pressure mapping [Adv. Mater. 2018, 30, e1801114; Adv. Mater. 2020, 32, e2001466; Sci. Adv. 2023, 9, eadg5152]. In this work, the active electrical signals were produced by the TENG sensing node upon external stimuli, and the signal processing circuit and wireless signal transmitter were driven by an external power source, enabling multifunctional sensing functions of the system (**revised Fig. 6a**).

Revised Fig. 6. (a) Schematic diagram of the actively signal progressing system of the UTE-skin for human-machine interfaces.

Revision to the manuscript:

On **page 14**, the sentence “As depicted in Fig. 5a, by actively responding to electrical signals, the UTE-skin serves as a self-powered human-machine interface in diverse system-level applications.” has been revised as “As depicted in Fig. 6a, in the system-level human-machine applications of the UTE-skin, the active electrical signals were produced by the TENG sensing node upon external stimuli, and the signal processing circuit and wireless signal transmitter were driven by an external power source, enabling multifunctional sensing functions of the system.”

Comment 2:

Crosstalk refers to the phenomenon where a signal being transmitted through one wire can affect nearby wires, leading to unexpected changes in the signals of those wires. a) The first type of crosstalk mainly occurs in a positive-negative cross-electrode array. When current flows through a sensor, it can pass through other sensors due to the absence of barriers, leading to crosstalk in the array of sensors. This paper uses a single-electrode model with a common ground, and there is no positive-negative electrode crossover. Therefore, this type of crosstalk does not exist. b) The second type of crosstalk occurs when there is a close spacing between two wires, leading to the presence of parasitic capacitance or coupled inductance. When a signal is transmitted through one wire, it can cause signal changes in the other wire due to the interaction between them. The paper indicates that the spacing between electrodes and wires is relatively large, with analysis shown in the diagrams to be around 1 cm or more. Therefore, this type of crosstalk also does not exist in this case. According to the paper, the signal interference mentioned primarily arises when there is no insulation provided by conductive carbon-black doping Ecoflex. In this case, the wires also act as electrodes and can generate frictional electric signals. Therefore, similar phenomena as shown in Figure 4e can occur. This type of interference may not be that significant in other designs, please comment on that.

Response 2:

The authors thank Reviewer #1 for providing the opinions regarding the crosstalk issues on the multiplexing sensors. In our work, a single-electrode model with a common ground was utilized instead of a positive-negative electrode crossover. Therefore, the crosstalk originating from the positive-negative cross-electrode arrays does not exist here. Meanwhile, the spacing between electrodes and wires was around 3 cm, resulting in negligible parasitic capacitance or coupled inductance, which can be validated in the results of **revised Fig. 4i-k**. As the electrode spacing gradually increased to 3 mm, the anti-interference index between two adjacent electrodes approached

96.0%, demonstrating the crosstalk occurring on closely spaced two electrodes was not notable in our design.

In the cases of single-electrode triboelectric sensing arrays [Adv. Mater. 2016, 28, 2896; Microsyst. Nanoeng. 2020, 6, 59; Nano Energy 2020, 76, 105047; Nano Energy 2021, 81, 105590], the “crosstalk” primarily refers to the electrical signal interferences between the sensing nodes and the internal circuits because both of the connecting wires and electrodes of single-electrode triboelectric sensing arrays can generate electrical signals when they are touched by human skin or interacting objects. This type of interference is a significantly fatal issue in single-electrode triboelectric sensing arrays, leading to the contradiction between the multiplexing design and the severe misrecognition [Adv. Mater. 2016, 28, 10024; Sci. Adv. 2017, 3, e1700015; Adv. Funct. Mater. 2019, 29, 1806379; Nano Energy 2022, 98, 107320]. This issue stems from intricate high-density electrode wiring design for multiplexing and large-area detection. When the high-density wiring is subjected to an external force/touch, an apparent electrical signal will be inevitably generated due to a combination of triboelectrification and electrostatic induction, which leads to the misrecognition of the true signal triggered by the sensing pixel. Such misrecognition can convey incorrect or misleading information about the physical stimulus, severely affecting sensing fidelity and hindering the application of TENG sensing matrices.

We used the word “crosstalk” because we referred previous reports [Adv. Mater. 2016, 28, 2896; Microsyst. Nanoeng. 2020, 6, 59; Nano Energy 2020, 76, 105047; Nano Energy 2021, 81, 105590]. Considering the reviewer’s opinion, we have changed the “crosstalk” to “misrecognition” to avoid any misunderstanding.

Revised Fig. 4. (i) Schematic diagram of the electrical signal tests of UTE-skin with two adjacent sensing panels spaced d apart. (j) Output voltage signals of the two adjacent sensing nodes with different electrode spacing by touching the left one. For a fair comparison, the tapping sites highlighted at the left panel were operated with a uniform contact force and area. (k) Plots of the statistical anti-interference index of UTE-skin sensory system relying on different spacings from panel (j), displaying the anti-interference abilities of tactile sensors at each spacing to improve the system-level design. Note that the index was extracted from the ratio of $(V_0 - V)/V_0$, where V and V_0 were the measured voltage values for each cell with possible disturbance and without disturbance, respectively. A curved arrow was utilized to guide the eyes.

Revision to the manuscript:

The result regarding the UTE-skin's resolution has been provided in **revised Fig. 4i-k**.

On **page 12**, “To enhance the integrability of sensory systems, the device resolution was further investigated by programming sensing node spacing from 0 to 5 mm.....Moreover, the extracted variation in sensing resolution versus spacing was believed to guide the operational effectiveness of the UTE-skin for use at the system level.” has been added.

Also, in the **Introduction part**, the sentence “However, a prevailing issue for TENG-based sensor arrays is the contradiction between the multiplexing design and severe misrecognitions.³⁰⁻³³” has been revised as “However, a prevailing and fatal issue for single-electrode triboelectric sensor arrays is the contradiction between the multiplexing design and the severe misrecognition, which primarily derives from the electrical signal interferences between the sensing nodes and the internal circuits.³⁰⁻³³”

Four relative references supporting the above sentence have been added to the revised manuscript as **Ref. 30** [*Adv. Mater.* **2016**, 28, 10024], **Ref. 31** [*Sci. Adv.* **2017**, 3, e1700015], **Ref. 32** [*Nano Energy* **2022**, 98, 107320], and **Ref. 33** [*Adv. Funct. Mater.* **2019**, 29, 1806379].

Comment 3:

According to Figure 1b, the "elastic triboelectric layer" and "triboelectric layer" seem to be considered as a single entity. The substrate is divided into two parts, and only the negative part of the triboelectric effect is mentioned, which is quite confusing.

Response 3:

The “elastic triboelectric layer” and “triboelectric layer” are a single entity and their separation in the schematics is for the clarity of the real fabrication process. The schematic illustration of the exploded view of the e-skin architecture involving elastic circuits, elastic electrodes, elastic triboelectric layer, and elastic shield layer was delivered in **revised Fig. 1b**. According to the fabrication procedure of the UTE-skin (**please refer to the first paragraph on page 5**), the pristine liquid Ecoflex was poured onto the patterned shielding layer (4×4 sensing array) as the triboelectric layer (thickness of 1.0 mm) and the Ecoflex composite was poured into its groove to construct elastic electrodes (*i.e.*, the sensing nodes) and connecting circuits. After casting a layer of Ecoflex elastomer as the substrate, peeling off the entire device from the acrylic mold, and turning it upside down, the UTE-skin with three-layer lamination (each layer was 1.0 mm) was ultimately completed. Accordingly, the top “elastic triboelectric layer” and “triboelectric layer” were finally cured as a single entity, functioning as the triboelectric layer in a 4×4 sensing array that created a continuous electrical output upon successive contact and separation with the human skin.

The bottom Ecoflex served as an ideal elastic substrate as well as encapsulating material that protects the elastic electrodes and connecting circuits from mechanical damage. It was also used as an underlying substance for placing onto other surfaces. In practice, there was no contact-and-separation between this side and the finger; therefore, it would not be triggered. As the

Ecoflex substrate and the elastic electrodes and circuits were eventually merged as the same layer, the “elastic substrate” component in the schematic diagram of the **original Fig. 1b** has been revised to avoid confusion.

Original Fig. 1. (b) Schematic showing the exploded view of the UTE-skin architecture involving elastomeric substrate, elastic circuits, elastic electrodes, elastic shield layer, and elastic triboelectric layer.

Revised Fig. 1. (b) The exploded view of the UTE-skin architecture involving elastic circuits, elastic electrodes, elastic triboelectric layer, and elastic shielding layer.

Revision to the manuscript:

The legend of **Fig. 1b** “Schematic showing the exploded view of the e-skin architecture involving elastomeric substrate, elastic circuits, elastic electrodes, elastic shield layer, and elastic triboelectric layer.” has been revised as “**The exploded view of the UTE-skin architecture involving elastic circuits, elastic electrodes, elastic triboelectric layer, and elastic shielding layer.**”

On **page 5**, the sentences “**In this work, the top Ecoflex layer functioned as the triboelectric layer in a 4×4 sensing array, which can create a continuous electrical output upon successive contact and separation with the human skin. Meanwhile, the bottom Ecoflex served as an ideal elastic substrate as well as encapsulating material that protected the elastic electrodes and connecting circuits from**

mechanical damage. It was also used as an underlying substance for placing onto other surfaces. In practice, there was no contact-and-separation between this side and the finger; therefore, it would not be triggered.” have been added.

Comment 4:

The paper mentions that Figure 1e and 1f exhibit numerous efficient conducting pathways. However, without additional information or access to the complete paper, it is challenging to provide specific details on how these conducting pathways are achieved.

Response 4:

The authors thank Reviewer #1’s constructive comment on the formation of conducting pathways in the Ecoflex-carbon black composites, and we agree with this viewpoint. The incorporation of nano/micro conductive fillers with elastomeric binders has been intensively studied for stretchable conductors owing to their high conductivity and stability [*Chem. Soc. Rev.* **2019**, *48*, 2946]. The conduction mechanism of the stretchable composites relies on the embedded percolation of conductive fillers, with elastomeric binders filling the interspace of the conductive networks [*Nat. Commun.* **2023**, *14*, 7132]. In this work, the charge transport between carbon particles in the composite involves the conducting pathways arising from the percolation junctions of carbon particles. The main points that contribute to the conductive performance of the Ecoflex-carbon black composites can be ascribed to three aspects:

1. As shown in **revised Fig. 1e**, the carbon blacks can be well embedded in the Ecoflex and the conductive composite contained large quantities of percolated carbon particles arranged in a dense network, which contributed to the conducting pathways of the composite. Meanwhile, the interspace between carbon black particles enabled Ecoflex to penetrate the conducting networks and embed the particles. **Revised Fig. 1f** shows a cross-section SEM image of the composite, which also validated that the carbon blacks were embedded in the Ecoflex without delamination. Such morphological characteristics regarding conductive composites have also been validated for the previously reported carbon black and carbon nanotubes/Ecoflex composite [*ACS Nano* **2016**, *10*, 7973], AgNWs/Ecoflex composite [*Adv. Mater.* **2016**, *28*, 10024], Ag flakes/fluorine rubber composite [*Nat. Mater.* **2017**, *16*, 834] and coffee ground/Ecoflex composite [*Nano Energy* **2021**, *79*, 105405].
2. In this work, the impact of carbon black content on the degree of electrostatic interference screening of the shielding layer was investigated. In **Fig. 2g, h**, we show the V_{oc} and I_{sc} of the elastic circuit modules at various carbon black mass fractions in the Ecoflex composites. It reflects a sharp decrease in V_{oc} and I_{sc} values with increasing carbon black contents from 0 wt% (43 V and 280 nA) to 4.8 wt% (0.6 V and 20 nA), and then 6.9 wt% (0.3 V and 0.4 nA), as a result of the further reinforcement of the electrostatic shielding effect. Since charge transport between carbon particles involves particle-particle percolation junctions, carbon black with a 4.8 wt% content can provide more conducting pathways, thus facilitating the efficient transfer of electrostatic induction charges. Further, at a carbon black content of 6.9 wt%, V_{oc} and I_{sc} drop to near zero as all electrostatic-induced charges of the circuit are shielded by a highly conductive shielding layer with numerous conducting pathways. Thus, the electrical results in **Fig. 2g, h** can

reveal that the electrostatic interference screening of the shielding layer (6.9 wt% carbon black) benefited from the numerous conducting pathways.

- The effects of carbon black content on the resistance of the shielding layer were also systematically investigated, as shown in **revised Fig. 3j, k**. The resistance of the shielding layer changed slightly when the carbon black content increased from 1.4 wt% to 3.2 wt% ($\sim 10^{11} \Omega$) (**revised Fig. 3j**). As the carbon black content further increased, the average resistance decreased by four ($\sim 10^7 \Omega$, 4.8 wt%) and five ($\sim 10^6 \Omega$, 6.9 wt%) orders of magnitude. These rises in conductivity correlate with the degree of percolation of the carbon particles, which produces numerous conducting pathways. In addition, resistance changes in the composites with carbon black contents of 4.8 wt% and 6.9 wt% at varying uniaxial strains up to 100% are also shown in **revised Fig. 3k**. The composite with 4.8 wt% carbon black showed an abrupt resistance change at 100% strain as cracks generated at the particle-particle junctions deteriorated the conducting pathways. However, at a 6.9 wt% carbon black, the resistance witnessed a smaller degradation, as the excess carbon particles greatly enhanced cross-linking between carbon particles. Here, the shielding layer with high conductivity (6.9 wt% carbon black) was utilized to efficiently transport electrostatic charges and eliminate misrecognition interference among adjacent sensing elements and circuits. Therefore, the effect of carbon black content on the resistance variations of the Ecoflex composites can prove the existence of numerous conducting pathways in the composites with 6.9 wt% carbon blacks.

Revised Fig. 1. (e) Top-view and (f) cross-section SEM images of the conductive composite of carbon blacks and Ecoflex. Scale bar: 100 nm.

Fig. 2. Dependence of the output (g) voltage and (h) current of the elastic circuit on the mass loading fraction of carbon blacks in the shielding layer.

Revised Fig. 3. (j) Resistance-strain characteristics of the carbon-blacks-doped Ecoflex shielding layer with different carbon-black contents. (k) Dependence of the normalized resistance changes of the shielding layer with 4.8 and 6.9 wt% carbon blacks on various strains up to 100%.

Revision to the manuscript:

The magnitude top-view and cross-section SEM images of the conductive Ecoflex-carbon black composite have been presented in **revised Fig. 1e, f**.

On **page 6**, the sentence “The conductive composite of carbon black and Ecoflex exhibited well-dispersed nanoparticles, which led to numerous efficient conducting pathways.” has been revised as “As shown in Fig. 1e, carbon blacks can be well embedded in the Ecoflex, and the composite contained large quantities of percolated carbon particles, which created conducting pathways in the composite. Meanwhile, the voids between carbon black particles enabled Ecoflex to penetrate the conducting networks and embed the particles. Fig. 1f shows a cross-section SEM image of the composite, which also validated that the carbon blacks were embedded in the Ecoflex without delamination. Such morphological characteristics regarding conductive composites have also been validated for the previously reported carbon black and carbon nanotubes/Ecoflex composite,⁴⁵ AgNWs/Ecoflex composite,³⁰ Ag flakes/fluorine rubber,⁴³ and coffee ground/Ecoflex composite.⁴⁸”.

In addition, four relative references have been added as **Ref. 45** [*ACS Nano* **2016**, 10, 7973], **Ref. 30** [*Adv. Mater.* **2016**, 28, 10024], **Ref. 43** [*Nat. Mater.* **2017**, 16, 834] and **Ref. 48** [*Nano Energy* **2021**, 79, 105405] to support our discussions.

Comment 5:

Figure 2a and 2b are not clearly annotated, making it difficult to understand their content easily.

Response 5:

The authors thank Reviewer #1’s comment for raising the quality of **Fig. 2a, b**. To make it easier to understand the working mechanism of TENG operating at single-electrode mode without and with a shielding layer, the annotations of the triboelectric layer, electrode, shield layer, human skin, Ecoflex, and carbon black particle have been added in **revised Fig. 2a, b**.

Revised Fig. 2. Schematic illustration of the working mechanism of TENG operating at single-electrode mode (a) without and (b) with a shielding layer.

Comment 6:

In Section 2.2, the human skin is considered as a positively charged layer. Therefore, the presence of the robot in Figure 1c can be quite misleading.

Response 6:

We agree. To make a clear presentation, the schematic diagram of the cartoon version of human triboelectric skin has been provided in **revised Fig. 1c**. It indicated that UTE-skin enabled conformal contact with soft human skin for a vast range of epidermal haptic sensing functions.

In addition, the deformable UTE-skin also enabled robotic skins. As indicated in **Fig. S10**, stable and substantial electrical outputs can be generated by the UTE-skin when contacting metallic materials, such as aluminum (Al) (29 V, 107 nA) and copper (Cu) (25 V, 97 nA). The autonomous generation of electrical signals by UTE-skin in response to stimuli from metallic materials supported the feasibility of a robotic skin.

Revised Fig. 1. (c) The broad applications of the UTE-skin, including electronic skins, robotic skins, stretchable keyboards, smart insole, and smart gloves.

Fig. S10. Dependence of (a) V_{oc} and (b) I_{sc} of the sensing node of UTE-skin on different kinds of contact materials.

Revision to the manuscript:

On page 6, the sentence “Accordingly, it can potentially be developed for a vast range of deformable haptic sensing devices, including robot skins, stretchable keyboards, intelligent gloves, and smart insoles (Fig. 1c).” has been revised as “Accordingly, it can potentially be developed for a vast range of deformable haptic sensing devices, including electronic skins, robotic skins, stretchable keyboards, intelligent gloves, and smart insoles (Fig. 1c).”

On page 11, the relative description “It should be noted that stable and substantial electrical outputs can be generated by the UTE-skin when contacting metallic materials, such as Al (29 V, 107 nA) and Cu (25 V, 97 nA). The active electrical signals by UTE-skin in response to stimuli from metallic materials supported the feasibility of a robotic skin, as depicted in Fig. 1c.” has been added.

Comment 7:

Figure 3b and 3c display the potential and current information of three triboelectric sensors.

Response 7:

Fig. 3b, c represents the output voltage and current signals of the sensing nodes, elastic circuits, and grounded elastic circuits as a function of uniaxial strain. To evaluate the screening effect of the shielding layer systematically, two modules with dimensions of $5 \times 5 \text{ cm}^2$ were fabricated to separately represent the sensing node and elastic circuit of the UTE-skin. Fig. 3b, c demonstrates the importance of the shielding layer for the performance of the UTE-skin. As the strain increased to 100%, the V_{oc} of the **sensing node** slightly decreased from 43 V to 30 V, whereas I_{sc} decreased from 130 nA to 40 nA. Notably, **with the shielding layer**, the amplitude of the electrical output substantially diminishes when the elastic circuits are touched under the same tension. When the **shielding layer was further grounded**, V_{oc} and I_{sc} were nearly zero because all the triboelectric charges on the connected circuits were screened. In this case, the misidentification of the stretched circuits remains to be eliminated. Consequently, the reliability and robustness of the UTE-skin can be considerably reinforced.

Revised Fig. 3. (b) Voltage and (c) current signals of the sensing nodes, elastic circuits, and grounded elastic circuits as a function of uniaxial strain.

Revision to the manuscript:

According to Reviewer #1's suggestion, the legend of **Fig. 3b, c** "(b) Voltage and (c) current signals of the two modules including sensing nodes and elastic circuits as a function of uniaxial strain." has been revised as "(b) Voltage and (c) current signals of the sensing nodes, elastic circuits, and grounded elastic circuits as a function of uniaxial strain."

Comment 8:

Figure 3k demonstrates the good electrical properties of conductive Ecoflex. However, there is a problem with understanding this. Even if we assume that the conductivity remains constant during the stretching process, the cross-sectional area of the material would decrease as it is stretched. According to the principles of incompressible materials, the cross-sectional area would shrink to at least 50% of its original size, resulting in an increase in resistance by at least twice or more. Additionally, why does the initial resistance start below 1?

Response 8:

The authors thank Reviewer #1's comment. **Revised Fig. 3k** represented the **normalized resistance changes** ($\Delta R/R_0$) of the shielding layer with 4.8 and 6.9 wt% carbon blacks on various strains. Here, the normalized resistance change was defined as $\Delta R/R_0 = (R - R_0)/R_0$, where R_0 and R were the resistances of the shielding layer before and after the deformation, respectively. According to the Pouillet's law [*Soft Robot.* **2018**, *5*, 175], the resistance of an ideal single material conductor with a uniform cross-section as a function of its resistivity and dimensions: $R = \rho l/A$, where R is resistance, ρ is resistivity, l is length, and A is the cross-sectional area, respectively. For typical materials with a positive Poisson's ratio, applying tensile stresses results in an increase in length, a reduction in cross-sectional area, and thus an increase in resistance.

As depicted in **revised Fig. 3k**, the shielding layer with 4.8 wt% carbon black had an abrupt resistance change, with R/R_0 of 3.5 at 10% strain, 16.0 at 50% strain, and 441 at 100% strain, respectively. However, at 6.9 wt% carbon black contents, the R/R_0 was 1.6 at 10% strain, 2.0 at 50% strain, and 2.6 at 100% strain, respectively. The variations in the resistance of the shielding layer with 4.8 and 6.9 wt% carbon black contents both complied with Poisson's law. Furthermore, we anticipate that at 4.8 wt% carbon black, fractures that developed at particle-particle junctions

deteriorated the conducting pathways, which could be a source of sudden resistance changes upon deformation. As the mass fraction increased to 6.9 wt%, there was a smaller degradation in conductivity as the excessive carbon particles significantly strengthened the cross-linking, and the formation of fractures was dampened owing to the stiffening effect of the particles. Thus, the shielding layer requires high conductivity (6.9 wt% carbon black) to efficiently transport electrostatic charges and eliminate misrecognition interference among adjacent elements and circuits.

In addition, as shown in Fig. 3j, the initial resistance R_0 (0% strain) was about $10^7 \Omega$ and $10^6 \Omega$ for 4.8% and 6.9 wt% carbon black contents, respectively. As there was no resistance change at this stage, the initial $\Delta R/R_0$ (0% strain) was zero for samples with both 4.8% and 6.9 wt% carbon black contents. And, as shown in revised Fig. 3k, the initial values representing $\Delta R/R_0$ (10% strain) were about 2.5 and 0.6 for 4.8% and 6.9 wt% carbon black contents, respectively.

Fig. 3. (j) Resistance-strain characteristics of the carbon-blacks-doped Ecoflex shielding layer with different carbon-black contents.

Revised Fig. 3. (k) Dependence of the normalized resistance changes of the shielding layer with 4.8 and 6.9 wt% carbon blacks on various strains up to 100%.

Revision to the manuscript:

On page 10, the sentences “In addition, resistance changes in the shielding layers with carbon black contents of 4.8 wt% and 6.9 wt% at varying uniaxial strains up to 100% are also shown in Figure 3k.....As the mass fraction increased to 6.9 wt%, there was no degradation in conductivity because the excessive carbon particles significantly strengthened the cross-linking, and the formation of fractures was dampened owing to the stiffening effect of the particles.⁴⁴” have been revised as “**In addition, normalized resistance changes ($\Delta R/R_0$) in the shielding layers with carbon black contents of**

4.8 wt% and 6.9 wt% at varying uniaxial strains are also shown in Fig. 3k. Here, the normalized resistance change is defined as $\Delta R/R_0 = (R-R_0)/R_0$, where R_0 and R are the resistances of the shielding layer before and after the deformation, respectively.....As the mass fraction increased to 6.9 wt%, there was a smaller degradation in conductivity because the excessive carbon particles significantly strengthened the cross-linking, and the formation of fractures was dampened owing to the stiffening effect of the particles.⁵¹”

In addition, the relative legend “Dependence of the resistance of the shielding layer with 4.8 and 6.9 wt% carbon blacks on various strains up to 100%.” has been revised as “**Dependence of the normalized resistance changes of the shielding layer with 4.8 and 6.9 wt% carbon blacks on various strains up to 100%.**” in revised Fig. 3k.

Comment 9:

In Figure 3c and f, larger error bars exist in current signals in low-strain cases. Why? Will it affect the sensing result in practical applications?

Response 9:

The authors thank Reviewer #1’s question on the larger error bars that exist in current signals in low-strain cases. In this work, we characterized the electrical outputs triggered by applying a 0.4 KPa contact force driven by a motor and stretched the device while maintaining the same area of the impacting force. Since the Ecoflex was an elastic polymer material, the downward pressure and rebound time would not be identical, making it possible that after the first hit was ready for the second, the silicone rubber would not return to its original position immediately. This resulted in differences in the contact time between the external stimulus and the Ecoflex triboelectric layer within the first and second hits, as well as between the following hits. According to the equation $I(t) = dQ/dt$, where I was the output current, Q was the transferred charges, and t was the contact time [Adv. Energy Mater. 2016, 6, 1600505], the output current was determined by the contact time. Thus, during repeated tests, the collected current signals were subjected to relatively large errors due to the variations in contact time (as shown in revised Fig. 3c, f). The contact time dependence characteristic of output current can also be validated in revised Fig. S6. It showed that as the contact time increased from 0 to 3 s, the V_{oc} of the UTE-skin remained steady, whereas I_{sc} gradually decreased from 0.55 to 0.20 μ A, indicating the dominant effect of contact time on current signals.

Moreover, the composite electrodes rely on the particle-particle junctions of carbon blacks to conduct the electrical signals (as shown in revised Fig. 1e, f). When the stretching strain was relatively small (e.g. 10% to 40% strain), some particles were pulled apart and some were not yet, resulting in either smooth signal transmission or unsuccessful delivery, and therefore a relatively large error bar in the collected current signals upon repeated tests. When the stretching strain was relatively large (e.g. 60% to 100% strain), the deformation of the Ecoflex caused by the motor’s hit was relatively minimal, resulting in slightly different contact time between the external stimulus and the Ecoflex triboelectric layer in different tests. Thus, the error bars in the current signals were much lower than that of small stretching strains. Meanwhile, the fractures occurring at the article-particle

junctions of carbon blacks would cause minor impacts on the conductivity of the electrode at large strains, leading to smaller error bars in the current signals upon different tests.

In this work, the larger error bars that existed in current signals in low-strain cases would not affect the sensing results in practical applications. As shown in **revised Fig. 4, 5, and 6**, the sensing signals in the self-powered sensing applications (*e.g.* static and dynamic recognition of different letters, smart glove, smart insole, and human-machine interactions) were primarily derived from the output voltage signals upon different external stimuli. Since the output voltage primarily relies on the contact area instead of the contact time between the external stimulus and the triboelectric layer [*Adv. Energy Mater.* **2016**, *6*, 1600505; *Nat. Commun.* **2020**, *11*, 6186], it is relatively stable upon different motor's hitting times (as demonstrated in **revised Fig. 3b, e**). Therefore, the relatively high and stable voltage signals were readily utilized as the electrical sensing signals in the self-powered sensing applications, which guaranteed the stability, accuracy, and repeatability of the sensing platforms.

Revised Fig. 1. (e) Top-view and (f) cross-section SEM images of the conductive composite of carbon blacks and Ecoflex. Scale bar: 100 nm.

Revised Fig. 3. (b) Voltage and (c) current signals of the sensing nodes, elastic circuits, and grounded elastic circuits as a function of uniaxial strain.

Revised Fig. 3. (e) Voltage and (f) current signals of the sensing nodes, elastic circuits, and grounded elastic circuits as a function of biaxial strain.

Revised Fig. S6. Dependence of (a) V_{oc} and (b) I_{sc} of the sensing node on different contact time.

Revision to the manuscript:

On page 9, the sentence “In addition, the relatively large error bars existing in current signals in low-strain cases mainly resulted from the variations in contact time between external stimulus and Ecoflex triboelectric layer within different measurement times under identical stretching conditions (Supplementary Fig. 6). While, the relatively stable voltage signals can be utilized as the sensing electrical signals of the UTE-skin in the self-powered sensing applications, which ensured the accuracy, robustness, and repeatability of the sensing platforms.” has been added.

Comment 10:

In Figure 4, the UTE-skin is applied as a stretchable sensor network for mechanical signal sensing. The study of the response of stretchable sensor networks to different mechanical signal inputs is indeed crucial. The authors should include quantitative experiments and discussions on this part.

Response 10:

The authors thank Reviewer #1 for the suggestions and agree with the reviewer that detailed information on the electrical responses depending on different mechanical signal inputs should be provided. To investigate the capability of the UTE-skin as a self-powered mechanical signal sensor, we tested its output behavior on different mechanical signal inputs, including operating frequencies

and applied pressures. As indicated in **revised Fig. S11a**, V_{oc} of the UTE-skin displayed a steady output of up to 43 V with increasing operating frequencies from 0.25 Hz to 4 Hz. While the short contacting time (~ 2 s) at high frequency resulted in a fast charge flow, responsible for the elevated I_{sc} from 0.28 to 1.70 μA (**revised Fig. S11b**). **Revised Fig. S12** depicted the V_{oc} and I_{sc} of the UTE-skin under 0.25 Hz with different applied pressures from 0.2, 0.4, 0.8, and 1.2 KPa, which yielded 25.50, 45.01, 53.60, and 64.70 V, respectively. The relevant I_{sc} attained 0.50, 1.40, 1.80, and 2.02 μA , demonstrating that the electrical outputs were dependent on the applied pressures of external stimuli. Meanwhile, as shown in **revised Fig. S13**, V_{oc} and I_{sc} increased linearly as the applied pressures increased from 0.2 to 2.0 KPa, whereas the growth rate slowed with the applied pressures augmenting to 3.2 KPa.

The above results validated that the UTE-skin can produce substantial and robust electrical signals even under slight applied pressure with a low frequency, ensuring its capability and feasibility of self-powered mechanical signals sensing as a stretchable sensor network.

Revised Fig. S11. Dependence of (a) V_{oc} and (b) I_{sc} of the sensing node of UTE-skin on different operating frequencies.

Revised Fig. S12. Dependence of (a) V_{oc} and (b) I_{sc} of the sensing node of UTE-skin on different applied pressures.

Revised Fig. S13. Pressure-dependent (a) V_{oc} and (b) I_{sc} of the sensing node of UTE-skin.

Revision to the manuscript:

Based on Reviewer #1’s suggestion, the demonstration of the UTE-skin’s electrical responses upon different mechanical signal inputs has been added as **revised Fig. S11, S12, and S13**.

On page 11, the relative description “**To investigate the capability of the UTE-skin as a self-powered mechanical signal sensor, we studied its output behavior depending on various operating frequencies and applied pressures.....Meanwhile, UTE-skin can produce notable and robust electrical signals even under slight pressures with low frequencies, ensuring its feasibility as a stretchable sensor network for self-powered mechanical signal sensing.**” has been added.

Responses to Reviewer #2:

Reviewer #2: This manuscript reports a large-scale triboelectric nanogenerator (TENGs) tactile sensing array, named untethered triboelectric electronic skin (UTE-skin). This UTE-skin, featuring a 4x4 pixel array over a 25 cm x 25 cm area, combines self-powered sensing with exceptional stretchability, durability, and precise spatial accuracy. Made from flexible electronic materials like Ecoflex and carbon black-doped Ecoflex, it effectively minimizes crosstalk between sensors, ensuring stable and accurate readings even under extreme tensile strains. Applications include intelligent gloves for gesture recognition, smart insoles for gait analysis, and immersive training tools for human-machine interfaces. However, the reviewer recommends that this manuscript can be accepted in 'Nature Communications' with the following Minor Revision.

Response:

We thank Reviewer #2 very much for carefully reviewing our manuscript and providing us with constructive comments that are important to further improve the quality of the manuscript.

Comment 1:

While the presence of the carbon black-doped Ecoflex layer has been confirmed to improve misrecognition rates, additional explanation is needed to show how it compares with other TENG-based structures in terms of reduced misrecognition. Please demonstrate that the recognition rate of the UTE-skin is superior under the same conditions used in experiments reported in other studies.

Response 1:

The authors thank Reviewer #2 for the comments regarding the comparisons of the UTE-skin with other TENG-based structures in terms of reduced misrecognition. In this case, the misrecognition primarily referred to the electrical signal interferences between the sensing nodes and the internal circuits. The Ecoflex composite-based shielding layer utilized in this study can effectively reduce misrecognition from the internal wiring and guarantee low-level noise in the sensor array.

According to Reviewer #2's suggestion, we have made comparisons of the UTE-skin with other TENG-based structures in terms of reduced misrecognition. The misrecognition rate, R_m , was defined as $R_m = V_m/V_0$, where V_0 was the initial voltage value and V_m was the voltage value of misrecognition signals. As shown in **Table R1**, our proposed design delivered the highest recognition rate of approximately 99.8% (lowest misrecognition rate of 0.20%) than other TENG-based structures. This result demonstrated that the shielding layer can effectively reduce misrecognition between sensing nodes and internal connecting wiring, guaranteeing the stability, accuracy, and repeatability of the UTE-skin. More importantly, benefiting from the intrinsically and omnidirectionally stretchable shielding layer, the UTE-skin can simultaneously attain 100% uniaxial, 100% biaxial, and 400% isotropic stretchability.

Revised Table S1. Comparisons of UTE-skin with state-of-the-art TENG-based structures

Year	Device structures	Misrecognition rate/%	Recognition rate/%	Flexibility	Stretchability	Reference
2016	PDMS/electrode/EVA/top PET/bottom PET shield	40.0	60.0	√	×	Ref. ¹
2018	layer/PDMS/PMMA-PDMS/C arbon black-PDMS	12.5	87.5	√	×	Ref. ²
2018	shield layer/PET/ITO/silicone	25.0	75.0	√	uniaxial stretchability	Ref. ³
2020	tribolayer/electrode/shield layer/insulating layer/spacer	16.0	84.0	√	×	Ref. ⁴
2020	Al/Ecoflex/PVA/PEI	0.30	99.7	√	×	Ref. ⁵
2021	shield layer/PDMS/PI/Cu/PI/PDMS	5.60	94.4	√	30% uniaxial stretchability	Ref. ⁶
2023	Ecoflex/shield layer/Ecoflex-carbon blacks/Ecoflex	0.20	99.8	√	100% uniaxial, 100% biaxial, and 400% isotropic stretchability	This work

Revision to the manuscript:

Based on Reviewer #2's suggestion, the performance comparisons of UTE-skin with state-of-the-art TENG-based structures in terms of reduced misrecognition have been provided in **revised Table S1**.

On **page 8**, the sentence “**The misrecognition rate (R_m) in this study can be obtained by the formula $R_m = V_m/V_0$, where V_0 was the initial voltage value and V_m was the voltage value of misrecognition signals. As shown in Supplementary Table 1, the proposed UTE-skin design delivered the lowest misrecognition rate of 0.20% than state-of-the-art TENG-based structures.**” has been added.

The six relative references have been added to the revised supporting information as **Ref. S1** [*Adv. Mater.* **2016**, 28, 2896], **Ref. S2** [*Adv. Funct. Mater.* **2018**, 28, 1802989], **Ref. S3** [*Mater. Today* **2018**, 21, 216], **Ref. S4** [*Nano Energy* **2020**, 76, 105047], **Ref. S5** [*Microsyst. Nanoeng.* **2020**, 6, 59], and **Ref. S6** [*Nano Energy* **2021**, 81, 105590].

Comment 2:

In Fig. 4a, the author has implemented the recognition of contact letters with the UTE-skin. It would be more appropriate to demonstrate the usability of e-skin by showing how accurately it can recognize letters when writing in real-time with a finger.

Response 2:

We agree. According to the suggestion, we have carried out experiments on recognizing letters by the UTE-skin with and without a shielding layer when writing the letter “N” in real-time with a finger. As shown in revised Fig. 4c-e, when writing on the UTE-skin without a shielding layer, the electrode circuits generated noticeable interfering signals, which conveyed incorrect and misleading information about the physical stimulus, severely affecting recognition fidelity. However, with a top shielding layer, the movement trajectory of the finger can be clearly recognized without interference from neighboring circuits according to voltage signals from different sensing pixels in a time-sequential manner (revised Fig. 4f-h). This confirms that UTE-skin with a shielding layer can accurately recognize the motion trajectories of finger sliding, which offers great potential for self-powered real-time haptic sensing of external stimuli.

Revised Fig. 4. (c) Photograph of a large-area matrix of 4×4 pixelated UTE-skin without a shielding layer ($25 \times 25 \text{ cm}^2$). (d) Schematic of sliding mode and (e) output voltages of pixels in the moving trajectory of the finger writing the letter “N” on the surface of UTE-skin without a shielding layer. (f) Photograph of a 4×4 pixelated UTE-skin with a shielding layer ($25 \times 25 \text{ cm}^2$). (g) Schematic of sliding mode and (h) output voltages of pixels in the moving trajectory of the finger writing the letter “N” on the surface of UTE-skin with a shielding layer.

Revision to the manuscript:

According to Reviewer #2’s suggestion, revised Fig. 4c-h has been provided in the manuscript.

On page 12, the relative description “Further, experiments on recognizing the letter “N” by the UTE-skin when writing in real-time with a finger were performed. As shown in Fig. 4c-e, when writing on the UTE-skin without a shielding layer, the electrode circuits generated noticeable interfering signals, which conveyed incorrect and misleading information about the physical stimulus, severely affecting recognition fidelity. However, with a top shielding layer, the movement trajectory of the finger can be clearly recognized without interference from neighboring circuits according to voltage signals from different sensing pixels in a time-sequential manner (Fig. 4f-h). This confirms

that UTE-skin with a shielding layer can accurately recognize the motion trajectories of finger sliding, which offers great potential for self-powered real-time haptic sensing of external stimuli.” has been provided

Comment 3:

The successful implementation of a large-area UTE-skin is well demonstrated. However, to further showcase its high applicability as an e-skin, it's necessary to develop and test a smaller version of the UTE-skin. For example, although the demo in Fig. 5e is conducted on a large area, data is needed for the creation and implementation of an e-skin small enough to be attached to the back of a hand. Additionally, please provide recognition rates for different distances between pixels to demonstrate the resolution of the e-skin.

Response 3:

Agreed. According to the suggestion, a smaller UTE-skin with five sensing nodes ($1.5 \times 1.5 \text{ cm}^2$) was fabricated and attached to the back of a human hand, operating as a wireless music controller (**Revised Fig. 6d and Supplementary Movie 2**). Five multifunctional sensing nodes were used as the “Play”, “Next”, “Last”, “Volume up”, and “Volume down” keys, respectively. Touching the UTE-skin “Play” key can remotely play the music in the player and pressing the UTE-skin “Next” key can play the next song in the player. Such demonstration of a self-powered microcontroller suggests promising uses for the integrated UTE-skin sensing systems in the fields of human–machine interfaces, supporting further research aimed at smart robotics.

In addition, the UTE-skin’s resolution was further investigated by programming sensing node spacing from 0 to 5 mm (**Revised Fig. 4i, j**). Output voltage signals of the two adjacent sensing nodes with different electrode spacing could be obtained by touching the left one. The tapping sites highlighted at the left panel were operated with uniform contact pressure and area for a fair comparison. **Revised Fig. 4k** displayed the statistical anti-interference index of UTE-skin sensory system relying on different spacings. The index was extracted from the ratio of $(V_0 - V)/V_0$, where V and V_0 were the measured voltage values for each pixel with possible disturbance and without disturbance, respectively. Notably, as the electrode spacing gradually increased to 3 mm, the anti-interference index approached 96.0%, demonstrating the high resolution of the UTE-skin sensor array. It enabled large-area, conformal precise tactile recognition and pressure mapping applications of the UTE-skin. Moreover, the extracted variation in sensing resolution versus spacing was believed to guide the operational effectiveness of the UTE-skin for use at the system level.

Revised Fig. 6. (d) Demonstration of operating the “Play” and “Next” instructions *via* triggering self-powered UTE-skin.

Revised Fig. 4. (i) Schematic diagram of the electrical signal tests of UTE-skin with two adjacent sensing panels spaced d apart. (j) Output voltage signals of the two adjacent sensing nodes with different electrode spacing by touching the left one. For a fair comparison, the tapping sites highlighted at the left panel were operated with a uniform contact force and area. (k) Plots of the statistical anti-interference index of UTE-skin sensory system relying on different spacings from panel (j), displaying the anti-interference abilities of tactile sensors at each spacing to improve the system-level design. Note that the index was extracted from the ratio of $(V - V_0)/V_0$, where V and V_0 were the measured voltage values for each pixel with possible disturbance and without disturbance, respectively. A curved arrow was utilized to guide the eyes.

Revision to the manuscript:

The result of UTE-skin’s application with a smaller version has been provided in **revised Fig. 6d and Movie S2**.

On **page 14**, the relative elaboration “A wireless UTE-skin-based music controller attached on the back of the human hand was further demonstrated. Five multifunctional sensing nodes ($1.5 \times 1.5 \text{ cm}^2$) were used as the “Play”, “Next”, “Last”, “Volume up”, and “Volume down” keys, respectively. Lightly touching the second or the third sensors freely controlled the instructions of “Play” or “Next”

(Fig. 6d and Supplementary Movie 2).” has been added.

Also, the result of the UTE-skin’s resolution has been provided in **revised Fig. 4i-k**.

On **page 12**, the relative elaboration “To enhance the integrability of sensory systems, the device resolution was further investigated by programming sensing node spacing from 0 to 5 mm (Fig. 4i, j). Notably, as the electrode spacing gradually increased to 3 mm, the anti-interference index approached 96.0% (Fig. 4k), demonstrating the high resolution of the UTE-skin sensor array. It enabled large-area, conformal precise tactile recognition and pressure mapping applications of the UTE-skin. Moreover, the extracted variation in sensing resolution versus spacing was believed to guide the operational effectiveness of the UTE-skin for use at the system level.” has been added.

Comment 4:

Since the title refers to 'Self-powered triboelectric skin', it would be beneficial to add data showing the extent of the UTE-skin's performance as an energy harvester.

Response 4:

Agreed. To show the capability of the UTE-skin as an energy harvester, we investigated its electrical output behavior depending on different mechanical signal inputs, including operating frequencies, contact time, and applied pressures. As indicated in **revised Fig. S11a**, V_{oc} of the UTE-skin displayed a steady output of up to 43 V with increasing operating frequencies from 0.25 Hz to 4 Hz. While the short contacting time (~ 2 s) at high frequency resulted in a fast charge flow, responsible for the elevated I_{sc} from 0.28 to 1.70 μA (**revised Fig. S11b**). **Revised Fig. S12** depicted the V_{oc} and I_{sc} of the UTE-skin under 0.25 Hz with applied pressures from 0.2, 0.4, 0.8, and 1.2 KPa, which yielded 25.50, 45.01, 53.60, and 64.70 V, respectively. The relevant I_{sc} attained 0.50, 1.40, 1.80, and 2.02 μA , demonstrating that the electrical outputs were dependent on the applied pressures of external stimuli. Meanwhile, as shown in **revised Fig. S13**, V_{oc} and I_{sc} increased linearly as the applied pressures increased from 0.2 to 2.0 KPa, whereas the growth rate slowed with the applied pressures augmenting to 3.2 KPa.

With such high electrical performance, the UTE-skin was able to instantaneously power up over 40 LEDs in series by gentle touching (**revised Fig. S14 and Movie S1**). In addition, we also demonstrated that the gathered mechanical energy by UTE-skin can drive an electronic watch after charging for only ~ 200 s (**revised Fig. S15**).

The above results provide strong evidence that the UTE-skin can be used as an energy harvester, serving as an efficient power supply that satisfies differential operation requests from users.

Revised Fig. S11. Dependence of (a) V_{oc} and (b) I_{sc} of the sensing node of UTE-skin on different operating frequencies.

Revised Fig. S12. Dependence of (a) V_{oc} and (b) I_{sc} of the sensing node of UTE-skin on different applied pressures.

Revised Fig. S13. Force-dependent (a) V_{oc} and (b) I_{sc} of the sensing node of UTE-skin.

Revised Fig. S14. Photograph showing that 40 commercial green LEDs were lit up when the device was touched.

Revised Fig. S15. Demonstration of (a) collecting mechanical energy from touching to power an electronic watch and (b) corresponding real-time charge/discharge curve.

Revision to the manuscript:

The electrical responses upon different mechanical signal inputs have been added as **revised Fig. S11, S12, and S13.**

Demonstrations of the UTE-skin associated with actuating LED arrays and an electronic watch have been provided as **revised Fig. S14, Movie S1, and revised Fig. S15.**

On page 11, the relative description “**To investigate the capability of the UTE-skin as a self-powered mechanical signal sensor, we studied its output behavior depending on various operating frequencies and applied pressures.....These results confirmed that the UTE-skin can be used as an energy harvester, serving as an efficient power supply that satisfies differential operation requests from users. Meanwhile, UTE-skin can produce notable and robust electrical signals even under slight pressures with low frequencies, ensuring its feasibility as a stretchable sensor network for self-powered mechanical signal sensing.**” has been added.

Comment 5:

Several references can be added to help the readers to understand this manuscript.

-Science and technology of advanced materials 20 (2019), 758-773

-Nano Energy 95 (2022) 107056

-Nano Energy 56, 531-546 (2019)

-Adv. Funct. Mater. 2022, 2112241

Response 5:

The authors thank Reviewer #2 for the suggestion and for kindly providing us with valuable references. The four relative references have been added to the revised manuscript as **Ref. 25** [*Sci. Technol. Adv. Mater.* **2019**, 20, 758], **Ref. 3** [*Nano Energy* **2022**, 95, 107056], **Ref. 7** [*Nano Energy* **2019**, 56, 531], and **Ref. 4** [*Adv. Funct. Mater.* **2022**, 2112241].

REVIEWERS' COMMENTS

Reviewer #1 (Remarks to the Author):

The authors have made very comprehensive revision based on the reviewers' comments. All my concerns have been addressed. Of course, I feel the paper has been further improved. I recommend the publication of this work as is.

Reviewer #2 (Remarks to the Author):

The authors have effectively responded to my comments by conducting additional experiments in the revised manuscript. Consequently, I would recommend this manuscript for possible publication in Nature Communications.